# Q: Do large language models understand implicature? A: Do pigs fly?

## Abstract

Despite widespread use of LLMs as conversational agents, evaluations of performance fail to capture a crucial aspect of communication: interpreting language in context. Humans interpret language using beliefs and prior knowledge about the world. For example, we intuitively understand the response "I wore gloves" to the question "Did you leave fingerprints?" as meaning "No". To investigate whether LLMs have the ability to make this type of inference, known as an *implicature*, we design a simple task and evaluate widely used state-of-the-art models. We find that, despite only evaluating on utterances that require a binary inference (yes or no), most perform close to random. Models adapted to be "aligned with human intent" perform much better, but still show a significant gap with human performance. We present our findings as the starting point for further research into evaluating how LLMs interpret language in context and to drive the development of more pragmatic and useful models of human discourse.

## 1 Introduction

> User: "Have you seen my phone?"
> InstructGPT: "Yes, I have seen your phone."

InstructGPT's response[1] is a perfectly fine answer to the question, but a human might answer differently. They might respond "it's in your bag," bypassing the obvious follow-up question ("where is it?"). Giving such a helpful and efficient answer is an example of pragmatic language usage that goes beyond the semantic meaning of utterances. Meaning is not only determined by a combination of words, but also context, beliefs, and social institutions (Wittgenstein, 1953; Grice, 1975; Huang, 2017). Consider another exchange where Esther asks her friend Juan "Can you come to my party on Friday?" and Juan responds "I have to work.". We resolve Juan's response into a decline by using the contextual commonsense knowledge that having to work on a Friday night precludes attendance. Both these exchanges contain an *implicature*—utterances that convey something other than their literal meaning[2]. Implicatures illustrate how context contributes to meaning; distinguishing writing and speaking from communicating (Green, 1996). We cannot fully understand utterances without understanding their implications, nor can a computational model. Indeed, the term "communication" presupposes the speaker's implications are understood by the addressee. Although communication encompasses much more than implicatures, such as assertives and other illocutionary acts, we view implicature understanding as a necessary condition for communicating with humans. Being able to resolve seemingly completely novel implicatures and—more broadly—engage in pragmatic understanding constitutes an essential and ubiquitous aspect of our every day usage of language.

Large language models (LLMs) have demonstrated remarkable ability on a variety of downstream tasks such as planning (Huang et al., 2022a), commonsense reasoning (Kojima et al., 2022), information retrieval (Lewis et al., 2020; Kim et al., 2022) and code completion (Austin et al., 2021; Biderman & Raff, 2022), to name just a few. When finetuned with human feedback, LLMs obtain higher ratings on desiderata like helpfulness (Ouyang et al., 2022; Bai et al., 2022), and are proposed as conversational agents (Thoppilan et al., 2022). Despite the widespread use and deploy-

---

[1] Appendix A contains details on how this answer was obtained from InstructGPT-3.
[2] In Appendix B we present a comprehensive introduction to implicature.

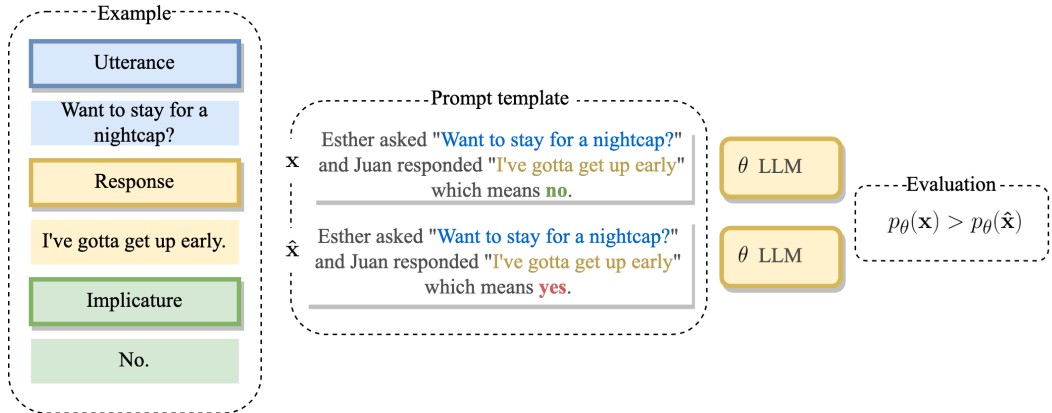

Figure 1: A schematic depiction of the protocol we propose to evaluate whether language models can interpret language in context. Each example in the test set gets wrapped in templates and transformed into an *incoherent* example by swapping "yes" and "no". The model is said to understand the implicature if it assigns a higher likelihood to the coherent text than the incoherent text.

ment of LLMs as conversational agents, there has been limited evaluation of their ability to navigate contextual commonsense knowledge.

This raises an important question: *to what extent do large language models understand conversational implicature?* To answer this question we use a publicly available dataset of conversational implicatures and propose an evaluation protocol on top of it (Figure 1). We evaluate a range of state-of-the-art models that can be categorised into four distinct groups; base LLMs (like OPT (Zhang et al., 2022)), instructable LLMs finetuned on downstream tasks (like Flan-T5 (Chung et al., 2022)), LLMs finetuned on conversational data (like BlenderBot (Ng et al., 2019)), and instructable LLMs finetuned with an unknown method (i.e. the latest versions of OpenAI's InstructGPT-3 series[3]). We evaluate both zero-shot and test whether performance improves by presenting in-context examples (few-shot evaluation). Our results suggest that implicature resolution is a very challenging task for LLMs. Most models obtain around 60% accuracy on the test set, whereas humans obtain 86% and random performance is 50%. InstructGPT-3 consistently outperforms other models across almost all model sizes considered, but even here zero-shot evaluation leaves a gap of 14% with the average human. In-context prompting can shrink this gap to 6% for the best of OpenAI's models. However, it does not help much for other models; at 30-shot they still all perform worse than instructGPT-3 does at zero-shot. We do a comprehensive error analysis by manually grouping the test examples into categories and uncover that the performance increase for the largest models seems driven by the simplest examples in the dataset that require no context to be resolved. For these examples the conventional meaning of the words entails a proposition, e.g. "some people came to the party" implying "not all people came". When isolating the best model's performance on implicatures that do require commonsense knowledge to be resolved (like the one in Figure 1), the gap between zero-shot and the human average becomes 24%, and the gap between few-shot and the human average becomes 9%. Furthermore, scaling analysis shows that most of the model classes we evaluate do not exhibit increased performance when scaled up. Based on this result, we hypothesise it is unlikely further scaling alone will lead to significant improvements.

The main contributions of this work are as follows i) we motivate implicature understanding as a crucial aspect of communication that is currently missing from evaluations of LLMs, ii) we design an implicature resolution task and propose a comprehensive evaluation protocol on which we evaluate both humans and LLMs to find that it poses a significant challenge for state-of-the-art LLMs, and (iii) we perform a comprehensive error analysis and identify opportunities for future work.

---

[3]The method is unpublished and might differ from the original instructGPT (Ouyang et al., 2022).

## 2 RELATED WORK

LLMs have demonstrated remarkable performance on tasks for which they were not explicitly trained (Brown et al., 2020). Building on the hypothesis that these abilities arise due to implicit multitask learning (Radford et al., 2019), the recent works of Sanh et al. (2022) and Wei et al. (2022) explicitly train LLMs in a supervised multitask fashion, leading to models that are better zero-shot learners with fewer parameters. Besides rapidly saturating language understanding benchmarks (Kiela et al., 2021), these advancements make LLMs beneficial foundations for agents performing a plethora of tasks (Adolphs et al., 2022; Reed et al., 2022). The trend towards using these models as agents brings along with it increased urgency for alignment with human values (Kenton et al., 2021). However, larger models trained with next-word prediction are generally more toxic and unhelpful (Gehman et al., 2020; Bender et al., 2021; Lin et al., 2022). Recent work mitigates this with approaches like prompting and finetuning on human-annotated outputs (Askell et al., 2021; Ouyang et al., 2022; Thoppilan et al., 2022). The produced models are more aligned on desiderata such as informativeness when evaluated by dedicated benchmarks and humans. We argue, however, that there is still something missing in these benchmarks. What is helpful and informative, as Kasirzadeh & Gabriel (2022) also point out, depends on the context in which a conversation is held. Consequently, any application of language models that requires communicating with humans will rely on pragmatic communication skills—something that is not explicitly captured by the benchmarks used to evaluate the alignment of LLMs.

The standard set of benchmarks LLMs are further evaluated on covers tasks like question answering (Berant et al., 2013; Joshi et al., 2017; Kwiatkowski et al., 2019), language completion (Levesque et al., 2012; Paperno et al., 2016; Mostafazadeh et al., 2016; Zellers et al., 2019; Sakaguchi et al., 2021), common-sense reasoning (Mihaylov et al., 2018; Clark et al., 2018; Bisk et al., 2020; Bhakthavatsalam et al., 2021), reading comprehension (Lai et al., 2017; Choi et al., 2018; Reddy et al., 2019; Dua et al., 2019), natural language inference (Rajpurkar et al., 2018; Nie et al., 2020), and more (Wang et al., 2019; Srivastava et al., 2022). Even though implicature is one of the most important aspects of language pragmatics (Levinson, 1983), none of these benchmarks *explicitly* evaluate implicature understanding. Reddy et al. (2019) evaluate implicit coreference among other aspects of conversation. This may indirectly measure performance on implicatures. However, unlike our work, it fails to decouple performance on implicatures from other aspects of pragmatics. Zheng et al. (2021) are the first to fill this gap with a dataset of conversational implicatures, called GRICE. This is important pioneering work highlighting the difficulty of implicature for language models, but their evaluations require task-specific training. In contrast, our evaluation protocol is applicable out-of-the-box and is much more comprehensive, evaluating models up to 176 billion parameters and using in-context prompting. Additionally, Zheng et al. (2021) benchmark synthetic data whereas this work evaluates performance on naturally occurring implicatures (George & Mamidi, 2020). We believe this to be a better representation of the true distribution of implicatures in natural dialogue.

Critiques of language modelling benchmarks are widespread (Raji et al., 2021; Bender et al., 2021; Bender & Koller, 2020; Raji et al., 2022). These works question whether the evaluation protocols measure what researchers claim they do. In similar spirit to our work, Valmeekam et al. (2022) point out that despite the fact that many works claim to use LLMs to "plan" (Ahn et al., 2022; Shah et al., 2022; Huang et al., 2022b) they either do not evaluate whether LLMs can do planning or use limited benchmarks that cannot justify the claims being made. Valmeekam et al. (2022) introduce an extensive evaluation suite for planning and find that "GPT-3 is, as of right now, pretty ineffective in reasoning about actions and change."

## 3 THE EVALUATION PROTOCOL

In this section we outline the full evaluation protocol we use to answer the research question "To what extent do large language models understand conversational implicature?". We focus on simple binary implicatures that require inferring "yes" or "no" (like the one in Figure 1). As a proxy for "understanding", we say a model *understands* an utterance if it assigns higher likelihood to a coherent utterance than a similar but incoherent one, detailed below.

**Zero-shot evaluation**. Consider the example from the introduction packed into a single utterance:

> Esther asked "Can you come to my party on Friday?" and Juan responded "I have to work", which means no.

We can transform this example to be *incoherent* (in the sense that it will become pragmatically inconsistent with expected use) by replacing the word "no" with "yes":

> Esther asked "Can you come to my party on Friday?" and Juan responded "I have to work", which means yes.

If the model understands the implicature, it should assign higher likelihood to the first of the two sentences above, namely the most coherent one. Importantly, both sentences have exactly the same words except for the binary implicature "yes" or "no", making the assigned likelihood scores directly comparable. Formally, let the coherent prompt be $\mathbf{x}$ and the augmented, incoherent prompt be $\hat{\mathbf{x}}$. A model outputs a likelihood $p$ parameterized by weights $\theta$. We say a model pragmatically *understands* an example $\mathbf{x}$ when it assigns $p_\theta(\mathbf{x}) > p_\theta(\hat{\mathbf{x}})$. This is equivalent to evaluating whether the model assigns a higher likelihood to the correct continuation of the two options. Note that this is a more lenient evaluation protocol than sometimes used for language models, where models are evaluated on on their ability to generate the correct continuation, in this case "no". However, "no" is not the only coherent continuation here, and marginalising over all possible correct continuations is intractable. The more lenient evaluation does capture implicature understanding, because the choice of "no" versus "yes" is only determined by the resolution of the implicature.

We use a dataset of conversational implicatures curated by George & Mamidi (2020). It contains conversational implicatures that, like in Figure 1, are presented in utterance-response-implicature tuples. Of these, 718 are binary implicatures that we can convert into an incoherent sentence. We randomly sample 600 examples for the test set and keep the remaining 118 as a development set to improve implicature understanding after pretraining through in-context prompting or finetuning.

**Few-shot in-context evaluation**. We add $k$ examples of the task to the prompt, e.g. with $k = 2$:

> The following examples are coherent sentences:
>
> Esther asked "Have you found him yet?" and Juan responded "They're still looking", which means no.
>
> Esther asked "Are you having fun?" and Juan responded "Is the pope Catholic?", which means yes.
>
> Finish the following sentence:
>
> Esther asked "Can you come to my party on Friday?" and Juan responded "I have to work", which means no.

We evaluate the models' $k$-shot capabilities for $k \in \{1, 5, 10, 15, 30\}$ by randomly sampling $k$ examples from the development set for each test example. We opt for a random sampling approach in place of the predominant approach in prior work which leverages the same ordered set of $k$ prompts for each test example. This change in protocol allows us to control for two sources of randomness. Firstly, examples have different levels of informativeness. Secondly, recent work found that the order in which examples are presented matters (Lu et al., 2022). Ideally, to marginalise over these random factors, we would evaluate each test example with all permutations of $k$ examples from the development set. This requires $\frac{118!}{(118-k)!}$ evaluations for each test example, which is intractable. Instead, we estimate performance per test example by randomly sampling from the development set. In this way we control for some of the variance in performance, but avoid extra evaluations.

**Controlling for prompt sensitivity**. It has been shown language models are sensitive to the wording of the prompt (Efrat & Levy, 2020; Tan et al., 2021; Reynolds & McDonell, 2021a; Webson & Pavlick, 2021). To control for this factor of randomness we manually curate six different template prompts and measure performance across these different wordings. One of the templates has already been presented in the examples in this section, namely "Esther asked *<utterance>* and Juan

responded *<response>*, which means *<implicature>*". Another prompt template is: "Question: *<utterance>*, response: *<response>*, meaning: *<implicature>*". The former we call *natural* prompts and the latter *structured* prompts. Each group has three templates that only differ slightly in wording. This grouping allows us to look at the variance due to slight changes in wording as well as performance difference due to a completely different way of presenting the example. The full list of prompts can be found in Table 4. As Perez et al. (2021) point out, for the few-shot evaluation to be truly few-shot, we formulate these prompt templates before any evaluation is done and never use more than $k$ examples from the development set for a test example.

## 4    EXPERIMENTS

The set of large language model classes we evaluate can be grouped into four distinct categories: (1) base models (namely RoBERTa (Liu et al., 2019), BERT (Devlin et al., 2018), GPT-2 (Radford et al., 2019), EleutherAI (Wang & Komatsuzaki, 2021; Black et al., 2022), BLOOM (BigScience, 2022), OPT (Zhang et al., 2022), and GPT-3 (Brown et al., 2020)), (2) LLMs finetuned on dialogue (BlenderBot (Ng et al., 2019)), (3) instructable LLMs finetuned on downstream tasks (T0 (Sanh et al., 2022) and Flan-T5 (Chung et al., 2022)), and (4) instructable LLMs finetuned with an unknown method (OpenAI's API models). Each group contains one or more model classes for which we evaluate a range of model sizes. A detailed categorization of the models and the attributes we discuss in the results can be found in appendix D.[4] We make use of the OpenAI and Cohere APIs as well as the pretrained models in the transformers library (Wolf et al., 2020) and EleutherAI's framework to evaluate them (Gao et al., 2021). All code used for this paper can be found on GitHub[5] and the dataset is made publicly available on HuggingFace[6]. We separately treat zero-shot and few-shot in-context evaluation, discussing performance for different model sizes of each model class and the variance over the prompt templates. Additionally, we manually group the test examples into categories and analyse what type of examples are difficult for the models. We contrast the models' performance with human performance. To this end, each test example gets annotated by five humans. We split the test set in four and assign each annotator a subset, giving us twenty annotators in total. Details on the human experiment can be found in the Appendix E. Detailed performance broken down by model and prompt template can be found in Appendix F.5.

### 4.1    ZERO-SHOT EVALUATION

**The best performing model classes overall**. Table 1 shows the best zero-shot accuracy each model class achieved on the implicature task. The OpenAI models ("UNK FT") perform significantly better than any other. The best accuracy is achieved by InstructGPT-3-175B (i.e. text-davinci-001, a 175 billion parameter model[7]) at $72\% \pm 2.8$. This leaves a gap of 13.9% with human average performance. Text-davinci-002 comes second with a zero-shot accuracy of $70.6\% \pm 2.3$, but the difference with InstructGPT-3-175B is not significant. All models in the other groups obtain performance closer to random than to humans (between 53.4% by BlenderBot-2.7B and 63.3% by Flan-T5-780M), showing a gap of at least 23% with the average human. We hypothesise that instruction finetuning as done for OpenAI's API models is especially important for this task, but we do not know the method and cannot say anything about it. In Appendix F.1 we reframe the task such that models can contrast the coherent and incoherent prompt, but this did not improve performance. Moreover, in Appendix F.3 we go into the stochasticity due to the fact that OpenAI's and Cohere's models are behind an API. After running the zero-shot experiment ten times through each API we conclude there is some stochasticity, but it is too small to impact the conclusions.

**Sensitivity to prompt wording**. As detailed in Table 4, each example in the test set is wrapped in six different prompt templates. The standard deviation in Table 1 shows the estimated sensitivity to different prompt wording. The standard deviation ranges from 0.3 for BlenderBot to 7.0 for T0-11B when looking at all templates. This variation is often much smaller when separating the performance over structured and natural prompts. Cohere-52B and BLOOM-7B1 are better at naturally worded

---

[4]Note that there are several important aspects unknown for models behind APIs, like Cohere and OpenAI.

[5]Supplied in supplementary material.

[6]When anonymity period is over link will appear here.

[7]For all OpenAI's API models except text-davinci-002 the size is assumed to align with the GPT-3 paper. There is reasonable evidence for this to be true `https://blog.eleuther.ai/gpt3-model-sizes/`

Table 1: The zero-shot accuracy for the best performing model of each class. The largest model does not always perform the best (i.e. for EleutherAI, BLOOM, OPT, GPT-3, BlenderBot, and Flan-T5). Column "all templates" has the mean performance on all templates. The std is over prompt templates for the models and over annotators for humans. The rightmost two columns hold a breakdown into the mean performance on the templates of the groups "structured" and "natural" respectively.

| | Model | All templates | Structured | Natural |
|---|---|---|---|---|
| **Baselines and Toplines** | Random | | 50% | |
| | Human avg. | | 86.2% ± 2.3 | |
| **Base models** | BERT-110M | 54.8% ± 1.6 | 56.1% ± 1.0 | 53.4% ± 0.2 |
| | RoBERTa-125M | 55.6% ± 2.0 | 54.1% ± 0.9 | 57.1% ± 1.5 |
| | GPT2-354M | 55.1% ± 2.6 | 53.5% ± 0.2 | 56.8% ± 2.8 |
| | EleutherAI-2.7B | 59.2% ± 3.1 | 56.4% ± 1.9 | 62.0% ± 0.6 |
| | BLOOM-7B1 | 58.7% ± 4.0 | 55.1% ± 2.5 | 62.2% ± 1.0 |
| | OPT-30B | 61.5% ± 1.9 | 62.5% ± 1.9 | 60.4% ± 1.2 |
| | Cohere-52B | 58.5% ± 4.0 | 54.6% ± 1.0 | 62.4% ± 0.3 |
| | GPT-3-1.3B | 57.7% ± 3.1 | 55.1% ± 1.3 | 60.4% ± 1.8 |
| **Dialogue FT** | BlenderBot-2.7B | 53.4% ± 0.3 | 53.6% ± 0.3 | 53.3% ± 0.0 |
| **Multitask FT** | T0-11B | 55.6% ± 7.0 | 62.2% ± 3.4 | 49.0% ± 0.7 |
| | Flan-T5-780M | 63.3% ± 2.8 | 61.4% ± 2.7 | 65.2% ± 1.1 |
| **UNK FT** | InstructGPT-3-175B | **72.3% ± 2.8** | **73.1% ± 3.7** | **71.5% ± 1.1** |
| | text-davinci-002-? | 70.6% ± 2.3 | 72.7% ± 1.0 | 68.5% ± 0.8 |

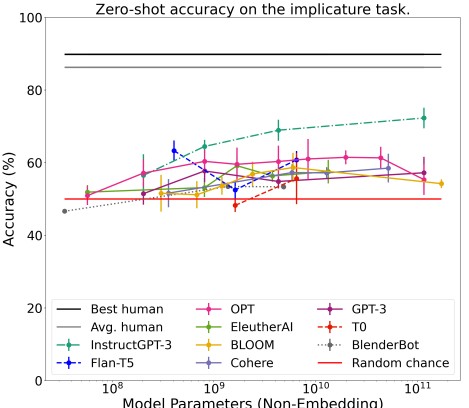
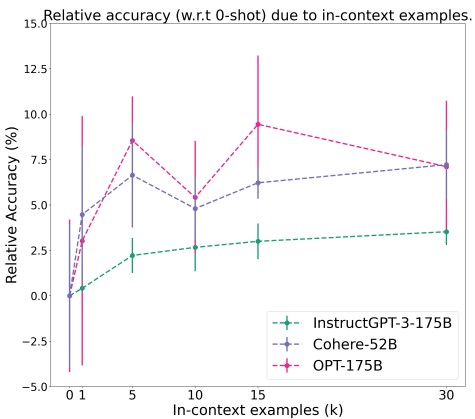

Figure 2: **Left:** The zero-shot accuracy for different sizes of the model classes. The error bars show standard deviation over prompt templates. OpenAI's instructable models perform better than most other models. For all models there is a significant gap between best accuracy and human accuracy. **Right:** Relative to zero-shot performance increase due to in-context examples, shown for the largest models of classes InstructGPT, Cohere, and OPT (note they are of a different size). The error bars show std. dev. over prompt templates. Performance increases strictly up to $k = 5$, and only slightly after. For OPT-175B there is a large variance over prompt templates.

prompts (template 2, 5, and 6 in Table 4), whereas OpenAI's models, T0-11B, and OPT-30B are better at structured prompts (template 1, 3, and 4 in Table 4). All in all, the sensitivity to prompt wording does not seem to be a problem for this task; the best and worst evaluations for each model do not change the fact that InstructGPT-3-175B perform best, but significantly worse than humans.

**The effect of scaling**. The left plot in Figure 2 shows the scaling laws we obtained from the model classes for which we know the number of non-embedding parameters. We again observe that OpenAI's instructable models perform significantly better than almost all other models on this task. Surprisingly, for many models the slope of the line is either near zero or decreasing. The only model

Table 2: An example from the dataset for each type of implicature found in the test set. The rightmost column shows the amount of that type we manually found in the test set.

| Type | Example Utterance | Example Response | Impl. | # |
|------|-------------------|------------------|-------|---|
| Generalised | You know all these people? | Some. | No. | 47 |
| Particularised | Want to stay for a nightcap? | I've gotta get up early. | No. | 88 |
| World knowledge | Did you leave fingerprints? | I wore gloves. | No. | 23 |
| Idiom | Would he fire me? | He's all bark and no bite. | No. | 42 |
| Rhetorical question | Can you drive that far? | Can fish swim? | Yes. | 11 |
| Other | - | - | - | 387 |

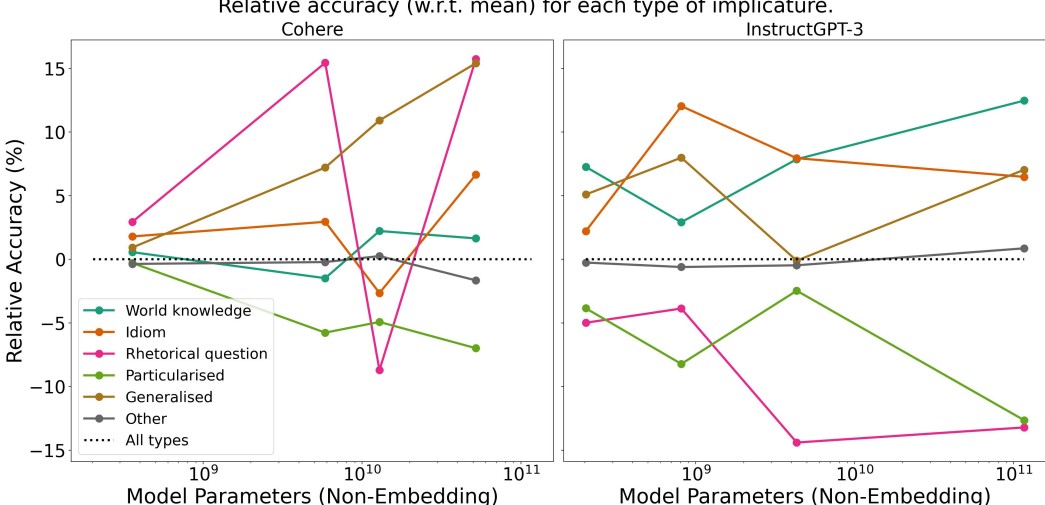

Figure 3: Relative accuracy (w.r.t. mean accuracy) for each example type for Cohere and InstructGPT-3. A point above the dotted line means the model gets that type right more often than the average performance on the test set. Particularised (context-heavy) examples are significantly more difficult than generalised (context-free) examples for both model classes. The type labels World knowledge, Idiom, and Rhetorical question do not show a significantly meaningful pattern.

classes for which the largest model performs best are Cohere, T0, and InstructGPT-3. For all other classes the largest model we tested obtains a worse performance than smaller versions. E.g. for GPT-3 the 1.3 billion parameter model performs better than the 175 billion parameter model. For Flan-T5 the smallest model of the class performs best.

**Breaking down performance per example type**. In Table 2 a taxonomy of the examples is shown, representing types of examples that occur frequently in the dataset. We manually labeled 213 examples of the 600 examples in the test set according to this taxonomy. The remaining 387 examples do not fall as clearly within a category and are grouped together as type *other*. *Generalised* implicatures require little or no context to be understood. They are the simplest type of example in the test set, and generally imply the same thing ("some" almost always implies "not all"). *Particularised* implicatures, by contrast, do require context to be resolved. For example, from Table 2, we need the context that it is undesirable to stay up late drinking when one has to get up early (see in Appendix B more generalised vs. particularised). The type *world knowledge* requires knowledge of the physical world to be resolved. From the example in Table 2; we need to know that you cannot leave fingerprints when wearing gloves to resolve this implicature. *Idiom* types contain an idiom or a metaphor that one needs to know or understand to resolve the implicature, and finally *Rhetorical question* types contain a question like "Is the Pope Catholic?", often requiring factual knowledge to be resolved. In Figure 3 the relative accuracy difference with the mean is shown for model classes Cohere and InstructGPT-3 (an absolute plot can be found in Appendix F.4). Generalised implicatures are relatively easier for almost all model sizes, and particularised implicatures are more difficult for all model sizes. In fact, for the largest models this difference becomes more

pronounced. Cohere-52B obtains a mean performance of 58.5% whereas for generalised examples it is 73.9% and for particularised examples it is 51.5%, which is close to random performance. For InstructGPT-3-175B the mean performance is 72.3%, whereas for generalised examples it is 79.3% and for particularised examples it is 59.7%. Humans also do worse on the particularised examples (83.2%), but the gap with the mean is smaller. Comparing the accuracy on these examples with humans uncovers a larger gap of 23.5% for InstructGPT-3-175B and 31.7% for Cohere-52B. The performance increase for larger models seems driven by the simple examples in the dataset that require no context to be resolved. We hypothesise that scaling up model size alone will not help with more complex implicature resolution. Moreover, as mentioned in Section 1, even though particularised implicatures do require context to be resolved, they are all implying a simple "yes" or "no". We conjecture that implicatures entailing several non-binary propositions are unlikely to be resolved by current SOTA language models.

**On prompting**. There is a narrative around large language models that if they fail a task, it might be that the prompt was not the right one (through works like Reynolds & Mc-Donell (2021b); Kojima et al. (2022)). The idea is that they can be prompted to simulate almost anything, if you set them up correctly. Because implicature resolution is a ubiquitous result of learning language, we hold the view that a model should be able to do this task if a prompt is given in coherent natural language. Nonetheless, in an additional effort to find the "let's think step-by-step" (Kojima et al., 2022) of zero-shot implicature resolution we try three more prompt templates. We evaluate a base large language model and the two best performing instructable models: GPT-3-175B, InstructGPT-3-175B, and text-davinci-002.

The prompts we use are taken from recent work that proposes a dialogue agent trained with human feedback (Glaese et al., 2022), but adapted to the task of implicature resolution. The full prompts are presented in Table 5 and Table 3 shows the results. The new templates do not improve performance for any of these models. The variance over the prompt templates for text-davinci-002 is very high, and the best prompt template of these three does achieve a slightly higher accuracy than the others: 74.5%. These results do not change the picture sketched so far. Of course, we will never claim a black swan does not exist, but given the breadth of our experiments we can conclude

Table 3: Zero-shot accuracy over three additional prompt templates for a base LLM and two instructable models.

| Model | Templates |
|---|---|
| GPT-3-175b | 59.2% $\pm$ 4.5 |
| InstructGPT-3-175b | 66.1% $\pm$ 3.2 |
| text-davinci-002-? | 67.7% $\pm$ 9.6 |

that using current LLMs to interpret language in context is non-trivial and advancements are needed.

## 4.2 FEW-SHOT IN-CONTEXT EVALUATION

**The effect of larger** k. We prompt the models with in-context examples from the development set to prime them for the task (detailed results in Appendix F.5). The highest accuracy we obtain is 80.6% $\pm$ 1.22, by text-davinci-002 for $k = 30$. This shrinks the gap with the average human to 5.6% and with the best human to 9.2%. Note that humans were tested zero-shot. When looking at the structured prompts, the accuracy is even slightly higher at 81.7% $\pm$ 0.9. The best performance due to in-context prompting of the other model groups is obtained by OPT-13B with 67.4% $\pm$ 2.1. Note that this is a worse accuracy than OpenAI's instructable models achieve zero-shot. The right plot in Figure 2 shows the relative performance increase due to prompting for the models InstructGPT-3-175B, Cohere-52B, and OPT-175B. In-context prompting boosts performance up to $k = 5$, for higher $k$ the performance barely increases. For OPT-175B there is a large variance in the effect. We stopped at $k = 30$ because the models' context windows could not handle more examples. Regardless, from Figure 2 it seems like larger $k$ would not increase performance significantly. In Appendix F.2 a small experiment is done to estimate the variance over prompt order for text-davinci-002, where the variance is again low enough to conclude this will not impact the results.

**The effect of in-context examples on sensitivity to prompt wording**. Figure 4 shows the relative performance increase due to in-context prompting broken down per prompt template. For InstructGPT-3-175B, most templates benefit similarly from more in-context examples, except for template 1. Perhaps surprisingly, we see that this template already achieves a performance of 76.5% at the zero-shot evaluation and does not improve much with few-shot prompting. For Cohere-52B and OPT-175B we see a clear grouping between the structured prompts (dashed lines) and natural prompts (dotted lines). Cohere struggles significantly more with the structured prompts than with

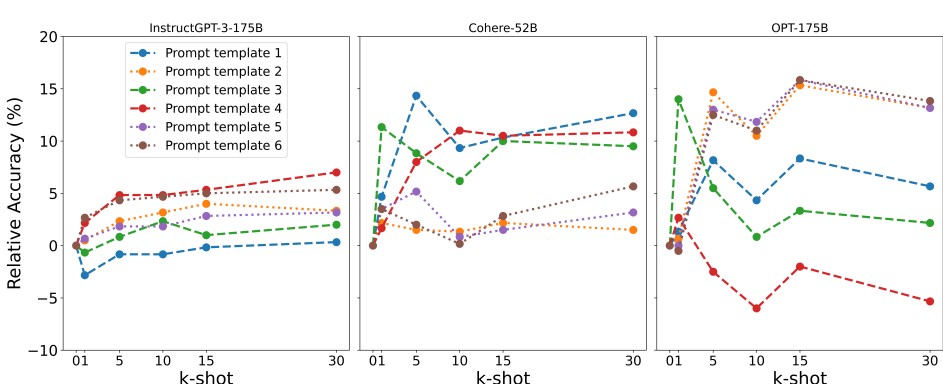

Figure 4: Relative performance increase over 0-shot due to in-context prompting. Structured prompt templates are dashed lines (1, 3, 4) and natural prompt templates dotted lines (2, 5, 6).

the natural prompts in the zero-shot evaluation, and few-shot prompting can mitigate that, lowering the standard deviation over prompt templates to 1.89 at $k = 30$ from 4 at $k = 0$. OPT benefits from prompting for the natural prompts, but not for the structured prompts.

**Breaking down performance per example type**. We observe again that the context-heavy examples are more difficult for the best performing model text-davinci-002 at $k = 30$. Recall that humans obtain a performance of 83.2% on the particularised examples. The model text-davinci-002 obtains a performance of 74.4% performance, leaving a gap of 8.8% with the average human.

## 5 CONCLUSION AND FUTURE WORK

Large language models have made remarkable progress on fluency and coherence in recent years. We argue however that a central aspect of language understanding is still missing. To understand language means to understand its pragmatics: its usage in context. We design a protocol that evaluates LLMs on binary implicature resolution and establish a significant gap with human understanding. The best performing models leave a gap of 13.9% with the average human in the zero-shot setting, and of 5.6% when $k = 30$. All other models obtain performance closer to random than to human performance. Model scaling plots and few-shot evaluations show increasing model size and prompt size is unlikely to close the gap. Moreover, when isolating performance on a context-heavy subset of the test set the gap becomes more pronounced. On context-heavy examples the gap with the average human for the best model is 23.5% in the zero-shot setting, and 8.8% when $k = 30$. We conjecture that a large part of the zero-shot performance increase for larger models is driven by simple examples in the dataset that require no context to be resolved. We further conjecture that the large difference in performance between OpenAI's text-davinci models and all other LLMs can be explained by the type of instruction finetuning they apply. However, without access to other instructable models (Thoppilan et al., 2022; Chowdhery et al., 2022) it is impossible to substantiate this hypothesis. Additionally, to substantiate the hypothesis that model size will not close the gap future work with larger model sizes is needed.

The type of implicatures we study is a simple type of conversational implicature that can be resolved to a yes or a no. This leaves ample room for the design of benchmarks with complex implicatures entailing more interesting propositions. Humans resolve much more complex propositions intuitively in conversation. For example, imagine Esther now asking "Can I use your stapler?" and Juan responding "Here's the key to my office.". Juan is implicating that (1) Esther can use the stapler, (2) the stapler is located in the office, and (3) the office is currently locked. Additionally, an interesting question for future work is for which accuracy models will be indistinguishable from humans. This could be answered with a type of Turing test in which a human must distinguish a LLM from another human by prompting both with a sequence of implicatures. We believe substantial work needs to be done to move beyond fluent text generation towards communication with autonomous agents and we hope this work will allow researchers to measure progress towards this goal.

## 6 REPRODUCIBILITY STATEMENT

We share all the data, human annotations, code used for the evaluations, and the raw results in the supplementary material. Additionally, in Appendix F.3 we estimate the variance due to stochasticity in the API's of OpenAI and Cohere. Of course, if either OpenAI or Cohere decides to change the models behind the API, the results might look different. We publish the exact date and time each API was queried for the results in Appendix G. Finally, in Appendix F.2 we estimate the variance over the prompt order of the in-context examples.

## 7 ETHICS STATEMENT

In this work, we conduct a study with human subjects (see Appendix E for details). To get matched with participants, we used the platform Prolific. Prolific complies with ethical standards according to UK law (e.g. complying with the GDPR). We compensated participants with a UK living wage at 15 GBP an hour, which is 6 GBP an hour more than Prolific recommends at 9 GBP per hour. Implicature is an aspect of pragmatics, and pragmatic language impairments are universal in Autism Spectrum Disorder (ASD) (American Psychiatric Association, 2013). Difficulties in understanding scalar implicatures are claimed to be present in people with ASD (Volden, 2017), although the nature of the relation has proven hard to establish and has recently been debated (Katsos et al., 2011; Schaeken et al., 2018). For the purposes of this work, whether or not implicature understanding relates to ASD is not important. We took the following steps to make sure no sensitive data is collected or published. The human annotations we obtain are anonymous, related to a participant only by their Prolific ID for the purposes of compensation. In publishing the human annotations, we will not publish the Prolific ID of participants or anything else related to the participants. Additionally, we did not collect or request any personal or demographic characteristics of the participants apart from that they are all native English speakers.

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

## A  OPENER EXAMPLE WITH INSTRUCTGPT

The opener quote by InstructGPT-3 was obtained through the OpenAI playground for text-davinci-002. The model text-davinci-001 consistently generates better responses. The following prompt was given:

> User: "Have you seen my phone?"
> InstructGPT:

With temperatures $t = \{0, 0.7, 1\}$. All three of text-davinci-002's responses were similar to:

> User: "Have you seen my phone?"
> InstructGPT: "Yes, I have seen your phone."

The model text-davinci-001 consistently generates:

> User: "Have you seen my phone?"
> InstructGPT: "No I have not seen your phone."

We tried extending the prompt, which gave similar results for text-davinci-002.

> The following is a request from a user. InstructGPT is a helpful and friendly conversational agent that tries to assist its users.
> User: "Have you seen my phone?"
> InstructGPT: "Yes, I have seen your phone."

The same approach makes text-davinci-001 a bit more helpful:

> The following is a request from a user. InstructGPT is a helpful and friendly conversational agent that tries to assist its users.
> User: "Have you seen my phone?"
> InstructGPT: "I haven't seen your phone, what type of phone is it?"

This is just a small experiment to illustrate a point, which half of the time goes wrong, even when prompted to be a helpful assistant. Of course, InstructGPT-3 cannot see, so the only "truthful" response is no.

## B  BACKGROUND ON IMPLICATURE

The first influential consideration of implicature is Grice (1975). In his work, Grice continues the trend of moving away from purely logical accounts of language started by Wittgenstein (1921) by hypothesising implicatures arise in conversation when some mutually agreed upon maxims seem to be violated. For example, if we agree on only making relevant contributions to conversation,

Juan's response in the introduction seemingly violates this maxim—after all, he starts talking about work when Esther asks him about a party. However, because Juan agreed to be relevant he must be implying that having to work means he cannot come to the party. Grice contrasts conversational implicatures that arise through context with conventional implicatures. These are implicatures where the *conventional* meaning of the word determines what is implicated. An example given by Grice is the following sentence: "he is an Englishman; he is therefore brave.". Grice notes that this sentence does not literally state that an Englishman being brave is a direct consequence of him being English, but it's implied by the conventional meaning of the word 'therefore'.

Since then, issues with the Gricean cooperative principle have been pointed out by many (Levinson, 1983; Sperber & Wilson, 1986; Davis, 1998; Lepore & Stone, 2014). The most influential alternative theory is relevancy theory by Sperber & Wilson (1986). They do away with the cooperative principle and instead theorise implicatures arise because speakers try to produce utterances that are both as relevant as possible and require the least effort to process. Another point of contention is the incorporation of conventional implicatures on the pragmatics side. Bach (1999) argues that there is no such thing as conventional implicatures, and they are simply instances of something else. Based on a thorough treatment of what Grice calls conventional implicatures, Bach argues all examples of it can be filed under other concepts within semantics, like utterance modifiers (called "utterance modifiers" instead of "sentence modifiers" because they go against the semantic content of the rest of the sentence). Potts (2005) also argues that to explain conventional implicatures we can stay on semantic turf. Indeed, even Grice himself says conventional implicatures derive from the meaning of the words, not from conversational context. However, Potts does not claim conventional implicatures do not exist, but instead argues they arise by a combination of lexical meaning and novel ways of combining words—the latter being the well-known principle of compositionality, an important part of semantics, not of pragmatics. Potts provides us with an illuminating demarcation between conventional and conversational implicatures. Conventional implicatures are never negotiable by context, whereas conversational implicatures are context-dependent and can always be cancelled without causing incoherent discourse. Consider again the sentence "he is an Englishman; he is therefore brave." and the sentence "Eddie has three bicycles" (implicating that Eddie has exactly three bicycles and not more). The former sentence can not be cancelled by new context without contradiction, whereas for the latter, if we continue saying "In fact, Eddie has 10 bicycles, he is a bicycle junkie", we have cancelled the implicature. This demarcation clearly puts conventional implicatures on the semantic side, and conversational implicatures on the pragmatic side. Potts goes on by providing a formal theory for conventional implicatures.

In later work, Potts (2006) describes how pragmatic pressures interacting with context cause conversational implicature to arise. He shows how sensitive conversational implicatures are to small changes in the context. Novel information about a speaker's belief state might completely change what is implied. There are many more models of implicature that aim to explain how humans understand language in context. Most notably, Frank & Goodman (2012) formalise the view that speakers produce utterances that are helpful and not longer than necessary with a Bayesian model called the rational speech act (RSA). Many variants on the RSA framework have since been proposed. For example, Goodman & Frank (2016) extend it to handle nonliteral uses of language, like irony, and metaphor. In the context of computational models, prior work uses insights from pragmatics to show that the use of certain words can make a language model produce biased completions (Patel & Pavlick (2021), e.g. saying someone "claimed" something rather than "said" something), and inform bias and sentiment classifiers (Greene & Resnik, 2009; Recasens et al., 2013).

In this work, we focus on conversational implicatures and not on conventional implicatures. All conversational implicatures are negotiable by context, but the way they depend on context can be different. Grice (1975) identifies generalised conversational implicatures and particularised conversational implicatures. The former require little or no context to be resolved. For example, "some athletes smoke" can imply "not all athletes smoke", but might also imply "I do not know whether all athletes smoke" when it is a response to the question "do you know whether all athletes smoke?" (Davis, 2019). The latter only arise in certain contexts. For example, the response "I have an early morning" to the question "do you want to stay for a drink?".

## C    DETAILED PROMPT TEMPLATES

Table 4 contains the full prompt templates we used for the main evaluation and Table 5 contains the extra prompt templates.

Table 4: The six templates we wrap the test examples in to present to the models. Template 1, 3, and 4 are of the type *structured*, and 2, 5, and 6 of the type *natural*. Within the type of prompt template they only differ slightly in wording.

| # | Prompt template |
|---|---|
| 1 | Does the following response to the question imply yes or no?

question: *<utterance>*
response: *<response>*
implicature: *<implicature>* |
| 2 | Finish the following text:

Esther asked "*<utterance>*" and Juan responded "*<response>*", which means *<implicature>* |
| 3 | Is the implied meaning of the following response yes or no:

question: *<utterance>*
response: *<response>*
meaning: *<implicature>* |
| 4 | What is the intent of the following response, yes or no?

question: *<utterance>*
response: *<response>*
intent: *<implicature>* |
| 5 | Finish the following text:

Karen asked "*<utterance>*" and William responded "*<response>*", which means *<implicature>* |
| 6 | Finish the following text:

Bob asked "*<utterance>*" and Alice responded "*<response>*", which means *<implicature>* |

## D    MODEL CATEGORIZATION

Table 6 contains details on the model classes that are a part of each group of models we evaluate, along with their model sizes.

## E    HUMAN EVALUATION

The participants for the human evaluation in this paper were recruited using Prolific (`www.prolific.co`). The setup of the experiment is as follows. We divide the test set of 600 examples into four non-overlapping subsets of 150 examples. Each set of 150 examples was given to five unique annotators. This means each example in the test set is labeled five times by different people, and we have in total twenty annotators for the whole test set (five different ones for each of the four subsets). The only constraint for the annotators is that they are native English speakers. In Figure 5 the screen shown to potential participants on Prolific is shown. Participants are paid 15 pounds an hour, which was the living wage at the time of the experiment and more than the 12 dollars an hour Prolific recommends.

The 150 test examples are wrapped in prompt template 2 (see Table 4) and presented in a Google form. The reason to wrap all examples in prompt template 2, as opposed to a mixture of all six templates is that although models have been shown to be very sensitive to prompt wording, humans are less likely to perform differently for different prompt templates. All templates are coherent natural language that any native English speaker will understand. That said, to confirm this hypothesis

Table 5: The three additional templates we wrap the test examples in to present to the models, adapted from (Glaese et al., 2022).

| # | Prompt template |
|---|---|
| 7 | The following text shows an interaction between two humans called Esther and Juan. In the interaction, Esther will ask Juan a question, and Juan will give an answer that contains an implicature. An implicature is an utterance that means something other than the literal meaning of the words. The implicature of Juan's response is yes or no. You, the AI assistant, are asked to finish the text with yes or no. The task begins: 

 Esther asked "*\<utterance\>*" and Juan responded "*\<response\>*", which means *\<implicature\>* |
| 8 | The following text shows an interaction between two humans called Esther and Juan. In the interaction, Esther will ask Juan a question, and Juan will give an answer that has a meaning besides the literal meaning of the words. That meaning is either yes or no. You, the AI assistant, are asked to finish the text with the correct meaning, either yes or no. The task begins: 

 Esther asked "*\<utterance\>*" and Juan responded "*\<response\>*", which means *\<implicature\>* |
| 9 | The following text shows an interaction between two humans called Esther and Juan. In the interaction, Esther will ask Juan a question, and Juan will give an answer that has a meaning besides the literal meaning of the words. That meaning is either yes or no. You, a highly intelligent and knowledgeable AI assistant, are asked to finish the text with the correct meaning, either yes or no. The task begins: 

 Esther asked "*\<utterance\>*" and Juan responded "*\<response\>*", which means *\<implicature\>* |

Table 6: Model categorization for each of the models. UNK stands for unknown, FT for finetuning, MT for multitask, and DL for dialogue.

| Group | Model class | Model IDs | Model size | Instruct |
|---|---|---|---|---|
| Base | BERT | base uncased | 110M | No |
| | RoBERTa | base, large | 125M, 355M | No |
| | GPT-2 | GPT-2 medium, large, xl | 354M, 774M, 1.6B | No |
| | EleutherAI | GPT-J, GPT-NeoX | 6B, 20B | No |
| | BLOOM | - | 560M, 1B1, 3B, 7B1, 176B | No |
| | OPT | - | 125M, 350M, 1.3B, 13B, 30B, 66B, 175B | No |
| | Cohere | small, medium, large, XL | 409.3M, 6.067B, 13.12B, 52.4B | No |
| | GPT-3 | ada, babbage, curie, davinci | Est. 350M, 1.3B, 6.7B, 175B | No |
| DL FT | BlenderBot | - | 90M, 2.7B, 9.4B | No |
| MT FT | T0 | - | 3B, 11B | Yes |
| | Flan-T5 | - | 780M, 3B, 11B | Yes |
| UNK FT | InstructGPT-3 | ada, babbage, curie, davinci-1 | Est. 350M, 1.3B, 6.7B, 175B | Yes |
| | text-davinci-002 | - | Unknown | Yes |

future work should investigate the effect of different wordings on implicature resolution by humans. The participants are asked to choose the correct continuation, yes or no (see Figure 6a). As recommended by Prolific, we subject the participants to an attention test (see Figure 6b). At three random places in the form, we add a question that does not contain an implicature and obviously maps to "yes". In this way, if the participants fails at least two of these questions, we can conclude they were not paying attention and remove their answers from the result. In practice, this happened once and we decided to pay the participant regardless, but discard their results, which were close to random.

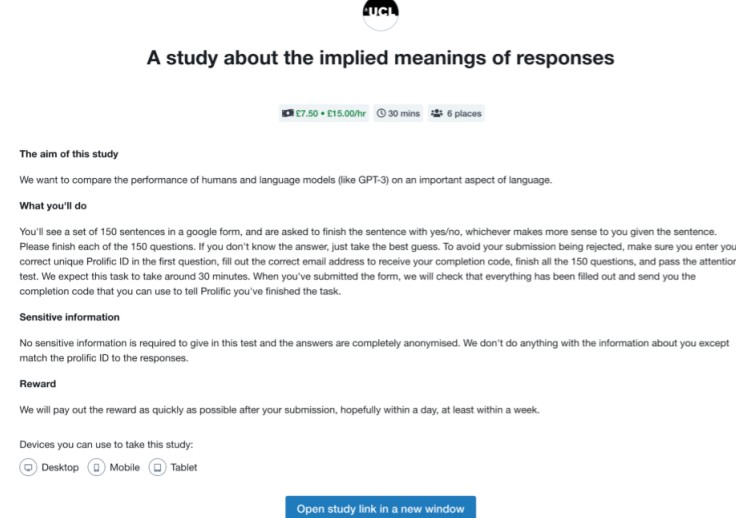

Figure 5: A screenshot of how the experiment is presented to potential annotators on Prolific (`www.prolific.co`).

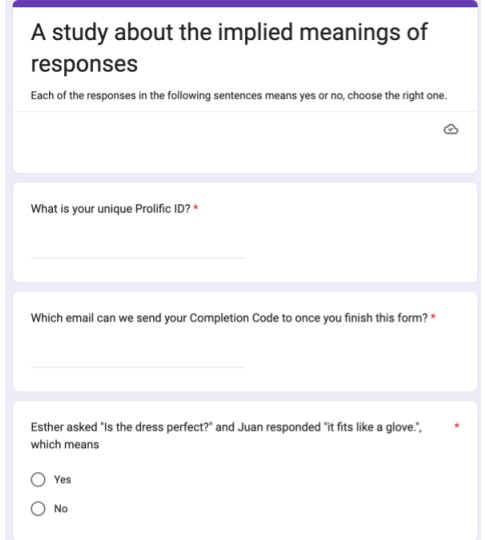

(a) The start of the Google form participants are asked to fill out for the human study.

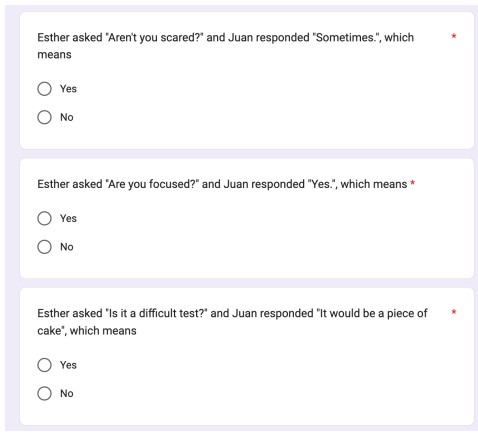

(b) Part of the Google form the participants are asked to fill out. The second question in this image is part of the attention test. Juan's response does not contain an implicature but simply gives away the correct answer.

Figure 6: Screenshots of the Google form participants fill out as part of the implicature study.

Table 7 shows the performance of each annotator on the subset they annotated. The average human performance across subsets and annotators is 86.2% ± 2.3, the best performance is 89.8% ± 2.2, and the worst performance is 83.5% ± 1.5. The column "IAA" shows the average Cohen's Kappa coefficient which is the pairwise inter-annotator agreement for each annotator per subset. All agreements are substantial according to the interpretation guidelines for Cohen's Kappa (between 0.61–0.80).

Table 7: The performance of the human annotators on the subsets of the test set. Subset 1 through 4 are non-overlapping and cover the whole test set. Annotator X for subset Y might be a different human than annotator X for subset Z. IAA is the average pairwise inter-annotator agreement (Cohen's kappa coefficient) between annotators per subset.

| Annotator | 1 | 2 | 3 | 4 | 5 | Mean | Best | Worst | IAA |
|-----------|-----|-----|-----|-----|-----|------|------|-------|-----|
| Subset 1 | 86.0% | 92.0% | 90.7% | 90.6% | 86.0% | 89.1% | 92.0% | 86.0% | 0.73 |
| Subset 2 | 84.7% | 83.3% | 87.3% | 86.0% | 86.0% | 85.5% | 87.3% | 83.3% | 0.64 |
| Subset 3 | 84.0% | 85.3% | 88.0% | 86.0% | 82.7% | 85.2% | 88.0% | 82.7% | 0.78 |
| Subset 4 | 85.3% | 82.7% | 84.0% | 82.0% | 92.0% | 85.2% | 92.0% | 82.0% | 0.71 |
| Total | - | - | - | - | - | 86.2% | 89.8% | 83.5% | 0.72 |
| Std | - | - | - | - | - | 2.3 | 2.2 | 1.5 | 0.1 |

# F ADDITIONAL RESULTS

## F.1 CONTRASTIVE EXPERIMENT

In this section we reframe the implicature resolution task to a contrastive one, allowing the model to contrast the coherent to the incoherent sentence in a single prompt.

**Contrastive task**. In the ranking task the model is required to assign higher likelihood to the coherent utterance than the incoherent one ($p_\theta(\mathbf{x}) > p_\theta(\hat{\mathbf{x}})$). In assigning a likelihood to $\mathbf{x}$, the model has no knowledge of $\hat{\mathbf{x}}$, and vice-versa. We hypothesize that the task might become easier if we reformulate it as a contrastive task. Consider the following prompt $\mathbf{p}$.

> Which of the following sentences is coherent:
>
> A: Esther asked "Can you come to my party on Friday?" and Juan responded "I have to work", which means no.
>
> B: Esther asked "Can you come to my party on Friday?" and Juan responded "I have to work", which means yes.
>
> Answer:

We can now evaluate the models' ability to understand which is the coherent sentence by evaluating whether it assigns $p_\theta(A \mid \mathbf{p}) > p_\theta(B \mid \mathbf{p})$. Note that this can again be framed in a ranking task of assigning a higher likelihood to the coherent prompt. If we finish the above prompt $\mathbf{p}$ by adding "A" to make a coherent prompt $\mathbf{x}$ and "B" to make an incoherent prompt $\hat{\mathbf{x}}$ we can again formulate the task by $p_\theta(\mathbf{x}) > p_\theta(\hat{\mathbf{x}})$. The difference is that within both the coherent and the incoherent prompt, the model can contrast the coherent and incoherent utterance to each other. We randomise the assignment of A and B to the utterances.

We do a small experiment with the contrastive task with the best performing model overall, OpenAI's text-davinci-002, for $k = \{0, 1, 5\}$. We use two prompt templates and for each template try three different multiple choice answers: A and B like above, one and two, or the full text of the answer. For the last option the coherent prompt $\mathbf{x}$ would look as follows:

> Which of the following sentences is coherent:
>
> A: Esther asked "Can you come to my party on Friday?" and Juan responded "I have to work", which means no.
>
> B: Esther asked "Can you come to my party on Friday?" and Juan responded "I have to work", which means yes.
>
> Answer: Esther asked "Can you come to my party on Friday?" and Juan responded "I have to work", which means no.

Table 8: Performance on the implicature task framed contrastively by OpenAI's text-davinci-002. The mean and standard deviation are reported over two different prompt templates (template 1 and 2).

| k | Non-contrastive | Rank one, two | Rank A, B | Rank full text |
|---|---|---|---|---|
| 0 | $71.3\% \pm 1.75$ | $53.9\% \pm 0.9$ | $59.3\% \pm 1.3$ | $48.9\% \pm 0.6$ |
| 1 | $76.1\% \pm 2.6$ | $59.4\% \pm 1.6$ | $63.2\% \pm 2.0$ | $66.9\% \pm 0.9$ |
| 5 | $80.5\% \pm 2.3$ | $61.4\% \pm 1.3$ | $64.0\% \pm 1.3$ | $67.9\% \pm 2.1$ |

In Table 8, perhaps surprisingly, we can see that the contrastive task is much more difficult than the original ranking task. For $k = 0$, the result is random except for the prompt where the multiple choice options are A and B. For $k = \{1, 5\}$ the full text ranking does best, but is still significantly worse than the original ranking setup. Because of these disappointing results, we did not evaluate the other models contrastively. Future work must establish whether the contrastive setup is worse across all model classes and sizes.

## F.2 VARIANCE OVER PROMPT ORDERING

As mentioned in Section 3, models are sensitive to the ordering of the $k$ examples in the prompt. Instead of marginalising over this random factor by evaluating all possible prompt orderings, we randomly sampled an ordered set of examples from the development set for each test example. Throughout experiments, we kept this randomly sampled order the same, meaning if you re-run the 5-shot evaluation you get exactly the same orderings. The reason for this is that we want evaluate each model equally. In this section we ask how the performance chances for the best performing model if we select another random order. We do this for the 5-shot evaluation, because the results show that adding more in-context examples barely helps performance.

Table 9: Variance over prompt ordering for 5-shot evaluation per prompt template (P.T.) for text-davinci-002

| Seed | P. T. 1 | P. T. 2 | P. T. 3 | P. T. 4 | P. T. 5 | P. T. 6 | Mean |
|---|---|---|---|---|---|---|---|
| 0 | 80.17 | 78.17 | 82.83 | 80.50 | 79.17 | 76.50 | 79.56 |
| 1 | 80.17 | 76.17 | 81.33 | 81.83 | 76.00 | 76.33 | 78.64 |
| 2 | 79.50 | 78.17 | 81.17 | 80.17 | 78.17 | 76.50 | 78.94 |
| mean | 79.94 | 77.50 | 81.78 | 80.83 | 77.78 | 76.44 | - |
| std | 0.31 | 0.94 | 0.75 | 0.72 | 1.32 | 0.08 | - |

Table 9 shows the results of this experiment. Some prompt templates seem to be more sensitive to prompt example ordering than others, but for none of them the variance is high enough to change any conclusions.

## F.3 VARIANCE OVER API RUNS

In this section we comment on the reproducibility of research done using APIs. Two of the model classes we evaluate have their models behind an API, meaning we do not have control over what happens to the prompt before the model processes it. We run the main evaluation, which is zero-shot, ten more times for the largest models of OpenAI and Cohere, text-davinci-002 and Cohere-52B. The results from this experiment are shown in Table 10 and 11. From this we can conclude that there is some stochasticity in the API that we have no control over, a bit more for OpenAI than for Cohere, but again we can be relatively confident that the conclusion will not be different because of it. The results from this work are therefore reproducible with access to the same models behind the API now. Unfortunately, when OpenAI or Cohere changes the models behind the API, these results are not exactly reproducible anymore.

For completeness, we add the timestamp that each result was obtained below (Appendix G).

Table 10: Results per prompt template (P.T.) for 10 different runs from text-davinci-002 for 0-shot evaluation.
Each evaluation has exactly the same text, so the variance in performance is due to API stochasticity.

| API-run | P. T. 1 | P. T. 2 | P. T. 3 | P. T. 4 | P. T. 5 | P. T. 6 | Mean |
|---------|---------|---------|---------|---------|---------|---------|------|
| 0 | 73.50 | 68.83 | 73.00 | 71.17 | 67.17 | 68.83 | 70.42 |
| 1 | 73.83 | 69.00 | 72.83 | 71.50 | 67.67 | 68.33 | 70.53 |
| 2 | 73.67 | 68.67 | 73.17 | 71.33 | 67.50 | 68.50 | 70.47 |
| 3 | 73.83 | 68.17 | 73.17 | 71.00 | 67.67 | 68.17 | 70.33 |
| 4 | 73.67 | 68.83 | 73.33 | 71.17 | 67.00 | 68.33 | 70.39 |
| 5 | 73.83 | 68.50 | 73.00 | 71.00 | 67.00 | 68.17 | 70.25 |
| 6 | 73.67 | 69.00 | 73.00 | 71.17 | 67.33 | 68.50 | 70.44 |
| 7 | 73.67 | 68.67 | 72.83 | 71.33 | 67.50 | 68.67 | 70.44 |
| 8 | 73.83 | 69.17 | 72.83 | 71.17 | 67.33 | 68.00 | 70.39 |
| 9 | 73.50 | 68.50 | 72.83 | 71.00 | 67.50 | 68.67 | 70.33 |
| 10 | 73.67 | 69.50 | 73.00 | 71.33 | 67.50 | 68.50 | 70.58 |
| mean | 73.70 | 68.80 | 73.00 | 71.20 | 67.38 | 68.42 | - |
| std | 0.12 | 0.35 | 0.16 | 0.16 | 0.23 | 0.24 | - |

Table 11: Results per prompt template (P.T.) for 10 different runs from Cohere-52B for 0-shot evaluation.
Each evaluation has exactly the same text, so the variance in performance is due to API stochasticity.

| API-run | P. T. 1 | P. T. 2 | P. T. 3 | P. T. 4 | P. T. 5 | P. T. 6 | Mean |
|---------|---------|---------|---------|---------|---------|---------|------|
| 0 | 56.00 | 62.67 | 54.33 | 54.00 | 62.17 | 62.17 | 58.56 |
| 1 | 56.00 | 62.83 | 54.33 | 54.00 | 62.33 | 62.33 | 58.64 |
| 2 | 56.00 | 62.83 | 54.33 | 54.00 | 62.17 | 62.33 | 58.61 |
| 3 | 56.00 | 62.83 | 54.33 | 54.00 | 62.17 | 62.33 | 58.61 |
| 4 | 55.83 | 62.67 | 54.33 | 54.00 | 62.17 | 62.33 | 58.56 |
| 5 | 56.00 | 62.83 | 54.33 | 54.00 | 62.17 | 62.17 | 58.58 |
| 6 | 56.00 | 62.83 | 54.33 | 54.00 | 62.17 | 62.17 | 58.58 |
| 7 | 56.00 | 62.67 | 54.33 | 54.00 | 62.33 | 62.17 | 58.58 |
| 8 | 56.00 | 62.83 | 54.33 | 54.00 | 62.00 | 62.33 | 58.58 |
| 9 | 56.00 | 62.83 | 54.00 | 53.83 | 62.17 | 62.17 | 58.50 |
| mean | 55.98 | 62.78 | 54.30 | 53.98 | 62.18 | 62.25 | - |
| std | 0.05 | 0.08 | 0.10 | 0.05 | 0.09 | 0.08 | - |

### F.4 ABSOLUTE TYPE LABEL ANALYSIS

Figure 7 shows the absolute accuracy for the type labels (from Section 4.1) that show a significant pattern; particularised and generalised. We observe increasing performance for generalised implicatures with scale, and decreasing or random performance for particularised implicatures.

### F.5 DETAILED RESULTS PER MODEL

This section contains the results used for the zero-shot and few-shot evaluation in the main text in Section 4, broken down per prompt template. See Table 12 until Table 58.

## G TIMESTAMPS API CALLS

For reproducibility purposes, Table 59 and 60 contain the dates and times the APIs from OpenAI and Cohere were queries for the results.

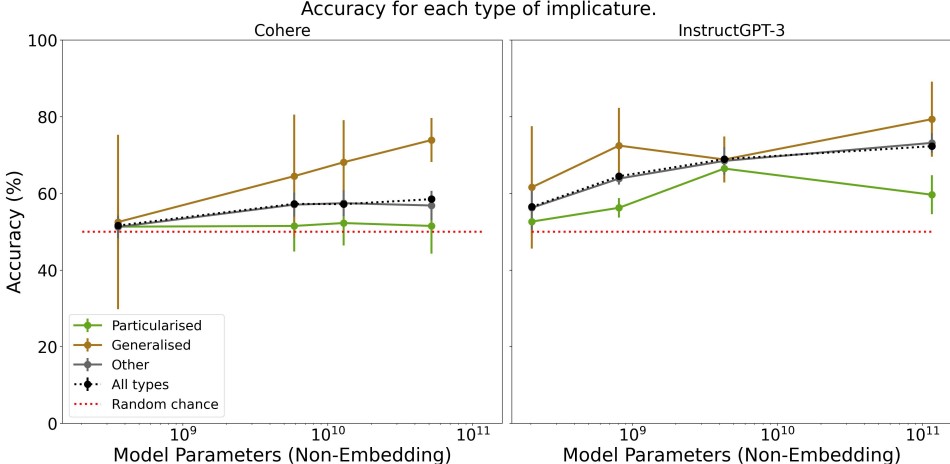

Figure 7: The absolute accuracy for each example type for model classes Cohere and InstructGPT-3. Particularised (context-heavy) examples are significantly more difficult than generalised (context-free) examples for both model classes. The type labels World knowledge, Idiom, and Rhetorical question do not show a significantly meaningful pattern and are left out of this plot. The error bars are standard deviation over prompt templates.

Table 12: Accuracy per prompt template for BERT-cased.

| Template | k = 0 | k = 1 | k = 5 | k = 10 | k = 15 | k = 30 |
|---|---|---|---|---|---|---|
| 1 | 47.3 | 48.8 | 50.5 | 49.8 | 46.7 | 46.7 |
| 2 | 46.8 | 50.3 | 45.5 | 50.2 | 46.7 | 46.5 |
| 3 | 57.3 | 51.5 | 50.0 | 50.0 | 47.0 | 46.7 |
| 4 | 48.8 | 51.0 | 49.5 | 48.5 | 46.8 | 46.7 |
| 5 | 46.7 | 50.3 | 44.5 | 47.7 | 46.7 | 46.7 |
| 6 | 46.7 | 50.3 | 45.8 | 47.8 | 46.8 | 46.7 |
| Mean | 48.9 | 50.4 | 47.6 | 49.0 | 46.8 | 46.7 |
| − std | 3.81 | 0.832 | 2.42 | 1.04 | 0.107 | 0.0745 |
| Structured | 51.1 | 50.4 | 50.0 | 49.4 | 46.8 | 46.7 |
| − std | 4.4 | 1.17 | 0.408 | 0.665 | 0.125 | 7.11e-15 |
| Natural | 46.7 | 50.3 | 45.3 | 48.6 | 46.7 | 46.6 |
| − std | 0.0471 | 7.11e-15 | 0.556 | 1.16 | 0.0471 | 0.0943 |

Table 13: Accuracy per prompt template for RoBERTa-base.

| Template | k = 0 | k = 1 | k = 5 | k = 10 | k = 15 | k = 30 |
|---|---|---|---|---|---|---|
| 1 | 54.0 | 55.8 | 58.0 | 58.7 | 58.3 | 57.8 |
| 2 | 56.5 | 50.5 | 52.0 | 55.8 | 56.0 | 54.2 |
| 3 | 53.0 | 56.8 | 56.8 | 61.3 | 59.5 | 58.8 |
| 4 | 55.2 | 56.0 | 58.7 | 59.8 | 56.8 | 57.2 |
| 5 | 55.7 | 50.3 | 52.3 | 54.8 | 55.5 | 53.0 |
| 6 | 59.2 | 50.3 | 54.2 | 55.8 | 55.7 | 55.3 |
| Mean | 55.6 | 53.3 | 55.3 | 57.7 | 57.0 | 56.1 |
| − std | 1.97 | 2.93 | 2.65 | 2.38 | 1.47 | 2.05 |
| Structured | 54.1 | 56.2 | 57.8 | 59.9 | 58.2 | 57.9 |
| − std | 0.899 | 0.432 | 0.785 | 1.07 | 1.1 | 0.66 |
| Natural | 57.1 | 50.4 | 52.8 | 55.5 | 55.7 | 54.2 |
| − std | 1.5 | 0.0943 | 0.974 | 0.471 | 0.205 | 0.939 |

Table 14: Accuracy per prompt template for RoBERTa-large.

| Template | k = 0 | k = 1 | k = 5 | k = 10 | k = 15 | k = 30 |
|---|---|---|---|---|---|---|
| 1 | 57.7 | 50.2 | 62.0 | 64.7 | 64.7 | 60.5 |
| 2 | 46.7 | 53.3 | 58.5 | 64.2 | 61.2 | 55.7 |
| 3 | 60.8 | 54.8 | 64.5 | 62.8 | 61.8 | 59.5 |
| 4 | 66.2 | 50.3 | 64.0 | 59.0 | 57.0 | 58.2 |
| 5 | 46.7 | 53.3 | 58.8 | 63.5 | 60.5 | 56.5 |
| 6 | 46.7 | 55.5 | 59.3 | 60.0 | 60.8 | 52.3 |
| Mean | 54.1 | 52.9 | 61.2 | 62.4 | 61.0 | 57.1 |
| – std | 7.84 | 2.03 | 2.45 | 2.13 | 2.26 | 2.7 |
| Structured | 61.6 | 51.8 | 63.5 | 62.2 | 61.2 | 59.4 |
| – std | 3.51 | 2.15 | 1.08 | 2.37 | 3.18 | 0.942 |
| Natural | 46.7 | 54.0 | 58.9 | 62.6 | 60.8 | 54.8 |
| – std | 7.11e-15 | 1.04 | 0.33 | 1.84 | 0.287 | 1.82 |

Table 15: Accuracy per prompt template for GPT-2-medium.

| Template | k = 0 | k = 1 | k = 5 | k = 10 | k = 15 | k = 30 |
|---|---|---|---|---|---|---|
| 1 | 53.2 | 53.7 | 54.0 | 53.8 | 53.8 | 55.0 |
| 2 | 52.8 | 53.7 | 55.8 | 57.2 | 60.3 | 57.2 |
| 3 | 53.7 | 54.0 | 52.5 | 56.5 | 55.8 | 55.3 |
| 4 | 53.5 | 55.7 | 53.3 | 55.8 | 55.5 | 54.3 |
| 5 | 59.2 | 54.3 | 56.7 | 57.7 | 60.7 | 58.8 |
| 6 | 58.3 | 54.8 | 55.7 | 57.7 | 61.7 | 57.8 |
| Mean | 55.1 | 54.4 | 54.7 | 56.4 | 58.0 | 56.4 |
| – std | 2.6 | 0.706 | 1.5 | 1.36 | 3.03 | 1.63 |
| Structured | 53.5 | 54.5 | 53.3 | 55.4 | 55.0 | 54.9 |
| – std | 0.205 | 0.881 | 0.613 | 1.14 | 0.881 | 0.419 |
| Natural | 56.8 | 54.3 | 56.1 | 57.5 | 60.9 | 57.9 |
| – std | 2.83 | 0.45 | 0.45 | 0.236 | 0.589 | 0.66 |

Table 16: Accuracy per prompt template for GPT-2-large.

| Template | k = 0 | k = 1 | k = 5 | k = 10 | k = 15 | k = 30 |
|---|---|---|---|---|---|---|
| 1 | 53.3 | 53.3 | 54.5 | 53.5 | 55.3 | 56.2 |
| 2 | 47.5 | 56.7 | 57.5 | 57.8 | 60.8 | 61.0 |
| 3 | 55.0 | 53.8 | 55.7 | 54.0 | 54.8 | 56.0 |
| 4 | 54.0 | 53.7 | 56.2 | 53.5 | 54.8 | 56.7 |
| 5 | 47.2 | 54.5 | 56.7 | 58.8 | 61.2 | 60.8 |
| 6 | 47.0 | 53.3 | 57.2 | 59.5 | 60.3 | 60.8 |
| Mean | 50.7 | 54.2 | 56.3 | 56.2 | 57.9 | 58.6 |
| – std | 3.47 | 1.18 | 1.0 | 2.57 | 2.92 | 2.29 |
| Structured | 54.1 | 53.6 | 55.5 | 53.7 | 55.0 | 56.3 |
| – std | 0.698 | 0.216 | 0.713 | 0.236 | 0.236 | 0.294 |
| Natural | 47.2 | 54.8 | 57.1 | 58.7 | 60.8 | 60.9 |
| – std | 0.205 | 1.41 | 0.33 | 0.698 | 0.368 | 0.0943 |

Table 17: Accuracy per prompt template for GPT-2-xl.

| Template | k = 0 | k = 1 | k = 5 | k = 10 | k = 15 | k = 30 |
|---|---|---|---|---|---|---|
| 1 | 53.2 | 53.3 | 57.0 | 54.5 | 54.7 | 56.2 |
| 2 | 48.7 | 61.3 | 57.3 | 63.7 | 62.0 | 60.5 |
| 3 | 55.0 | 55.2 | 59.5 | 59.0 | 58.0 | 60.7 |
| 4 | 54.2 | 54.3 | 56.0 | 54.5 | 54.3 | 56.3 |
| 5 | 48.0 | 59.7 | 58.3 | 60.8 | 62.7 | 61.7 |
| 6 | 48.5 | 60.8 | 58.0 | 61.8 | 61.5 | 61.5 |
| Mean | 51.3 | 57.4 | 57.7 | 59.1 | 58.9 | 59.5 |
| − std | 2.92 | 3.25 | 1.1 | 3.5 | 3.43 | 2.32 |
| Structured | 54.1 | 54.3 | 57.5 | 56.0 | 55.7 | 57.7 |
| − std | 0.736 | 0.776 | 1.47 | 2.12 | 1.66 | 2.1 |
| Natural | 48.4 | 60.6 | 57.9 | 62.1 | 62.1 | 61.2 |
| − std | 0.294 | 0.668 | 0.419 | 1.2 | 0.492 | 0.525 |

Table 18: Accuracy per prompt template for EleutherAI-125M.

| Template | k = 0 | k = 1 | k = 5 | k = 10 | k = 15 | k = 30 |
|---|---|---|---|---|---|---|
| 1 | 53.3 | 53.7 | 52.7 | 56.2 | 56.2 | 54.0 |
| 2 | 52.2 | 50.0 | 47.5 | 53.5 | 55.7 | 53.3 |
| 3 | 53.3 | 53.8 | 51.2 | 55.8 | 54.8 | 52.8 |
| 4 | 53.7 | 52.5 | 51.2 | 53.8 | 55.8 | 53.2 |
| 5 | 50.7 | 50.2 | 47.3 | 53.8 | 56.2 | 53.8 |
| 6 | 48.2 | 49.8 | 47.5 | 53.2 | 57.5 | 53.5 |
| Mean | 51.9 | 51.7 | 49.6 | 54.4 | 56.0 | 53.4 |
| − std | 1.93 | 1.72 | 2.19 | 1.17 | 0.806 | 0.394 |
| Structured | 53.4 | 53.3 | 51.7 | 55.3 | 55.6 | 53.3 |
| − std | 0.189 | 0.591 | 0.707 | 1.05 | 0.589 | 0.499 |
| Natural | 50.4 | 50.0 | 47.4 | 53.5 | 56.5 | 53.5 |
| − std | 1.65 | 0.163 | 0.0943 | 0.245 | 0.759 | 0.205 |

Table 19: Accuracy per prompt template for EleutherAI-1.3B.

| Template | k = 0 | k = 1 | k = 5 | k = 10 | k = 15 | k = 30 |
|---|---|---|---|---|---|---|
| 1 | 54.3 | 53.7 | 54.8 | 57.5 | 57.2 | 56.2 |
| 2 | 51.8 | 56.8 | 57.5 | 59.0 | 55.8 | 54.7 |
| 3 | 58.0 | 55.5 | 59.5 | 58.0 | 61.5 | 57.5 |
| 4 | 53.2 | 57.5 | 56.8 | 55.2 | 56.5 | 54.7 |
| 5 | 49.7 | 55.2 | 57.5 | 58.7 | 57.2 | 56.7 |
| 6 | 51.8 | 55.7 | 56.5 | 58.7 | 56.5 | 56.2 |
| Mean | 53.1 | 55.7 | 57.1 | 57.8 | 57.4 | 56.0 |
| − std | 2.59 | 1.21 | 1.4 | 1.29 | 1.87 | 1.02 |
| Structured | 55.2 | 55.6 | 57.0 | 56.9 | 58.4 | 56.1 |
| − std | 2.05 | 1.55 | 1.93 | 1.22 | 2.21 | 1.14 |
| Natural | 51.1 | 55.9 | 57.2 | 58.8 | 56.5 | 55.9 |
| − std | 0.99 | 0.668 | 0.471 | 0.141 | 0.572 | 0.85 |

Table 20: Accuracy per prompt template for EleutherAI-2.7B.

| Template | k = 0 | k = 1 | k = 5 | k = 10 | k = 15 | k = 30 |
|---|---|---|---|---|---|---|
| 1 | 54.0 | 52.8 | 58.2 | 57.8 | 59.5 | 56.7 |
| 2 | 62.0 | 56.2 | 57.7 | 55.8 | 57.8 | 57.7 |
| 3 | 58.7 | 60.0 | 58.8 | 59.2 | 57.8 | 57.8 |
| 4 | 56.5 | 54.2 | 57.5 | 56.2 | 57.5 | 55.5 |
| 5 | 62.7 | 54.7 | 58.7 | 55.7 | 57.3 | 57.8 |
| 6 | 61.2 | 55.2 | 57.3 | 57.5 | 58.5 | 58.7 |
| Mean | 59.2 | 55.5 | 58.0 | 57.0 | 58.1 | 57.4 |
| – std | 3.13 | 2.25 | 0.576 | 1.26 | 0.741 | 1.02 |
| Structured | 56.4 | 55.7 | 58.2 | 57.7 | 58.3 | 56.7 |
| – std | 1.92 | 3.12 | 0.531 | 1.23 | 0.881 | 0.939 |
| Natural | 62.0 | 55.4 | 57.9 | 56.3 | 57.9 | 58.1 |
| – std | 0.613 | 0.624 | 0.589 | 0.826 | 0.492 | 0.45 |

Table 21: Accuracy per prompt template for EleutherAI-6B.

| Template | k = 0 | k = 1 | k = 5 | k = 10 | k = 15 | k = 30 |
|---|---|---|---|---|---|---|
| 1 | 57.5 | 58.8 | 52.7 | 53.0 | 52.5 | 51.3 |
| 2 | 57.7 | 51.8 | 63.2 | 62.7 | 64.3 | 65.3 |
| 3 | 56.2 | 58.2 | 57.2 | 53.0 | 54.7 | 54.5 |
| 4 | 52.8 | 55.5 | 53.3 | 52.2 | 54.0 | 53.8 |
| 5 | 56.8 | 52.7 | 62.7 | 63.2 | 65.2 | 64.2 |
| 6 | 57.2 | 52.8 | 61.3 | 61.8 | 62.2 | 63.3 |
| Mean | 56.4 | 55.0 | 58.4 | 57.6 | 58.8 | 58.7 |
| – std | 1.67 | 2.75 | 4.28 | 4.94 | 5.2 | 5.65 |
| Structured | 55.5 | 57.5 | 54.4 | 52.7 | 53.7 | 53.2 |
| – std | 1.98 | 1.44 | 1.99 | 0.377 | 0.918 | 1.37 |
| Natural | 57.2 | 52.4 | 62.4 | 62.6 | 63.9 | 64.3 |
| – std | 0.368 | 0.45 | 0.804 | 0.579 | 1.26 | 0.818 |

Table 22: Accuracy per prompt template for EleutherAI-20B.

| Template | k = 0 | k = 1 | k = 5 | k = 10 | k = 15 | k = 30 |
|---|---|---|---|---|---|---|
| 1 | 53.0 | 58.0 | 55.3 | 54.3 | 52.8 | 54.3 |
| 2 | 61.3 | 54.2 | 65.8 | 63.3 | 65.0 | 60.3 |
| 3 | 54.3 | 58.3 | 58.5 | 56.7 | 55.3 | 52.0 |
| 4 | 56.2 | 58.2 | 55.3 | 57.2 | 57.0 | 58.7 |
| 5 | 59.0 | 53.0 | 66.7 | 62.8 | 65.0 | 59.2 |
| 6 | 61.3 | 53.5 | 65.2 | 61.7 | 64.0 | 59.7 |
| Mean | 57.5 | 55.9 | 61.1 | 59.3 | 59.9 | 57.4 |
| – std | 3.25 | 2.33 | 4.9 | 3.42 | 4.98 | 3.09 |
| Structured | 54.5 | 58.2 | 56.4 | 56.1 | 55.0 | 55.0 |
| – std | 1.31 | 0.125 | 1.51 | 1.27 | 1.72 | 2.78 |
| Natural | 60.5 | 53.6 | 65.9 | 62.6 | 64.7 | 59.7 |
| – std | 1.08 | 0.492 | 0.616 | 0.668 | 0.471 | 0.45 |

Table 23: Accuracy per prompt template for BLOOM-560M.

| Template | k = 0 | k = 1 | k = 5 | k = 10 | k = 15 | k = 30 |
|---|---|---|---|---|---|---|
| 1 | 54.3 | 54.2 | 53.5 | 53.8 | 53.8 | 53.5 |
| 2 | 46.7 | 56.3 | 54.0 | 54.8 | 56.0 | 55.3 |
| 3 | 58.8 | 53.3 | 53.8 | 53.3 | 54.5 | 54.0 |
| 4 | 56.3 | 54.8 | 53.5 | 54.8 | 52.7 | 56.7 |
| 5 | 46.7 | 54.3 | 53.7 | 55.3 | 56.3 | 55.5 |
| 6 | 46.7 | 56.0 | 54.0 | 55.2 | 56.7 | 55.0 |
| Mean | 51.6 | 54.8 | 53.8 | 54.5 | 55.0 | 55.0 |
| – std | 5.05 | 1.04 | 0.206 | 0.734 | 1.45 | 1.04 |
| Structured | 56.5 | 54.1 | 53.6 | 54.0 | 53.7 | 54.7 |
| – std | 1.84 | 0.616 | 0.141 | 0.624 | 0.741 | 1.41 |
| Natural | 46.7 | 55.5 | 53.9 | 55.1 | 56.3 | 55.3 |
| – std | 7.11e-15 | 0.881 | 0.141 | 0.216 | 0.287 | 0.205 |

Table 24: Accuracy per prompt template for BLOOM-1B1.

| Template | k = 0 | k = 1 | k = 5 | k = 10 | k = 15 | k = 30 |
|---|---|---|---|---|---|---|
| 1 | 53.3 | 53.5 | 56.2 | 54.2 | 55.2 | 54.5 |
| 2 | 49.0 | 51.5 | 58.2 | 59.8 | 58.8 | 60.8 |
| 3 | 57.2 | 54.2 | 55.8 | 54.0 | 55.5 | 50.8 |
| 4 | 53.3 | 54.0 | 54.2 | 53.3 | 55.7 | 55.8 |
| 5 | 47.3 | 51.2 | 59.8 | 61.3 | 60.2 | 60.0 |
| 6 | 46.8 | 51.0 | 60.2 | 61.2 | 60.2 | 59.3 |
| Mean | 51.2 | 52.6 | 57.4 | 57.3 | 57.6 | 56.9 |
| – std | 3.75 | 1.36 | 2.18 | 3.51 | 2.19 | 3.53 |
| Structured | 54.6 | 53.9 | 55.4 | 53.8 | 55.5 | 53.7 |
| – std | 1.84 | 0.294 | 0.864 | 0.386 | 0.205 | 2.12 |
| Natural | 47.7 | 51.2 | 59.4 | 60.8 | 59.7 | 60.0 |
| – std | 0.942 | 0.205 | 0.864 | 0.685 | 0.66 | 0.613 |

Table 25: Accuracy per prompt template for BLOOM-1B7.

| Template | k = 0 | k = 1 | k = 5 | k = 10 | k = 15 | k = 30 |
|---|---|---|---|---|---|---|
| 1 | 53.5 | 54.7 | 53.8 | 54.0 | 55.7 | 56.5 |
| 2 | 57.7 | 52.2 | 56.3 | 55.5 | 55.8 | 52.0 |
| 3 | 54.7 | 53.2 | 53.8 | 51.0 | 54.5 | 54.0 |
| 4 | 54.5 | 53.8 | 54.5 | 51.2 | 55.5 | 50.3 |
| 5 | 50.0 | 51.2 | 54.3 | 53.2 | 54.7 | 50.0 |
| 6 | 51.3 | 51.8 | 53.8 | 54.0 | 54.7 | 50.8 |
| Mean | 53.6 | 52.8 | 54.4 | 53.1 | 55.1 | 52.3 |
| – std | 2.49 | 1.2 | 0.886 | 1.6 | 0.528 | 2.31 |
| Structured | 54.2 | 53.9 | 54.0 | 52.1 | 55.2 | 53.6 |
| – std | 0.525 | 0.616 | 0.33 | 1.37 | 0.525 | 2.55 |
| Natural | 53.0 | 51.7 | 54.8 | 54.2 | 55.1 | 50.9 |
| – std | 3.37 | 0.411 | 1.08 | 0.953 | 0.519 | 0.822 |

Table 26: Accuracy per prompt template for BLOOM-3B.

| Template | k = 0 | k = 1 | k = 5 | k = 10 | k = 15 | k = 30 |
|---|---|---|---|---|---|---|
| 1 | 53.0 | 54.0 | 56.8 | 59.5 | 60.0 | 58.2 |
| 2 | 62.5 | 58.0 | 58.2 | 59.7 | 57.5 | 60.0 |
| 3 | 53.5 | 54.0 | 57.2 | 58.7 | 59.2 | 58.2 |
| 4 | 54.8 | 55.3 | 55.7 | 59.0 | 58.2 | 55.8 |
| 5 | 58.5 | 57.5 | 58.0 | 59.7 | 58.8 | 60.2 |
| 6 | 59.0 | 56.8 | 57.3 | 59.8 | 58.5 | 59.5 |
| Mean | 56.9 | 55.9 | 57.2 | 59.4 | 58.7 | 58.6 |
| − std | 3.4 | 1.6 | 0.823 | 0.408 | 0.783 | 1.5 |
| Structured | 53.8 | 54.4 | 56.6 | 59.1 | 59.1 | 57.4 |
| − std | 0.759 | 0.613 | 0.634 | 0.33 | 0.736 | 1.13 |
| Natural | 60.0 | 57.4 | 57.8 | 59.7 | 58.3 | 59.9 |
| − std | 1.78 | 0.492 | 0.386 | 0.0471 | 0.556 | 0.294 |

Table 27: Accuracy per prompt template for BLOOM-7B1.

| Template | k = 0 | k = 1 | k = 5 | k = 10 | k = 15 | k = 30 |
|---|---|---|---|---|---|---|
| 1 | 53.2 | 55.2 | 55.2 | 52.0 | 53.0 | 52.7 |
| 2 | 61.2 | 59.0 | 53.7 | 58.3 | 58.8 | 61.7 |
| 3 | 58.7 | 53.3 | 53.0 | 53.3 | 53.0 | 52.8 |
| 4 | 53.5 | 53.5 | 55.2 | 52.8 | 54.3 | 53.5 |
| 5 | 62.0 | 61.0 | 55.3 | 60.3 | 58.5 | 62.5 |
| 6 | 63.5 | 60.0 | 54.7 | 59.8 | 56.3 | 62.5 |
| Mean | 58.7 | 57.0 | 54.5 | 56.1 | 55.7 | 57.6 |
| − std | 4.03 | 3.11 | 0.871 | 3.46 | 2.39 | 4.63 |
| Structured | 55.1 | 54.0 | 54.5 | 52.7 | 53.4 | 53.0 |
| − std | 2.52 | 0.852 | 1.04 | 0.535 | 0.613 | 0.356 |
| Natural | 62.2 | 60.0 | 54.6 | 59.5 | 57.9 | 62.2 |
| − std | 0.953 | 0.816 | 0.66 | 0.85 | 1.11 | 0.377 |

Table 28: Accuracy per prompt template for BLOOM-176B.

| Template | k = 0 | k = 1 | k = 5 | k = 10 | k = 15 | k = 30 |
|---|---|---|---|---|---|---|
| 1 | 53.8 | 58.8 | 58.5 | 57.7 | 55.7 | 56.7 |
| 2 | 55.8 | 60.8 | 68.0 | 65.7 | 64.2 | 62.7 |
| 3 | 53.5 | 66.7 | 69.3 | 71.8 | 71.7 | 69.8 |
| 4 | 54.3 | 59.8 | 64.8 | 62.2 | 60.7 | 61.3 |
| 5 | 52.3 | 61.3 | 66.2 | 61.8 | 58.8 | 57.5 |
| 6 | 55.5 | 59.2 | 65.7 | 61.7 | 60.3 | 58.3 |
| Mean | 54.2 | 61.1 | 65.4 | 63.5 | 61.9 | 61.1 |
| − std | 1.19 | 2.65 | 3.43 | 4.38 | 5.06 | 4.44 |
| Structured | 53.9 | 61.8 | 64.2 | 63.9 | 62.7 | 62.6 |
| − std | 0.33 | 3.51 | 4.43 | 5.88 | 6.68 | 5.43 |
| Natural | 54.5 | 60.4 | 66.6 | 63.1 | 61.1 | 59.5 |
| − std | 1.58 | 0.896 | 0.988 | 1.86 | 2.28 | 2.29 |

Table 29: Accuracy per prompt template for OPT-125M.

| Template | k = 0 | k = 1 | k = 5 | k = 10 | k = 15 | k = 30 |
|----------|-------|-------|-------|--------|--------|--------|
| 1 | 53.3 | 55.2 | 54.0 | 55.2 | 54.2 | 55.0 |
| 2 | 49.5 | 50.5 | 47.5 | 52.7 | 50.5 | 48.2 |
| 3 | 53.5 | 55.5 | 53.0 | 55.0 | 53.7 | 56.0 |
| 4 | 53.3 | 54.5 | 54.2 | 53.8 | 54.3 | 53.8 |
| 5 | 48.5 | 50.5 | 46.3 | 50.7 | 49.5 | 48.0 |
| 6 | 47.3 | 50.2 | 46.3 | 50.0 | 49.0 | 48.0 |
| Mean | 50.9 | 52.7 | 50.2 | 52.9 | 51.9 | 51.5 |
| − std | 2.55 | 2.35 | 3.56 | 1.99 | 2.25 | 3.49 |
| Structured | 53.4 | 55.1 | 53.7 | 54.7 | 54.1 | 54.9 |
| − std | 0.0943 | 0.419 | 0.525 | 0.618 | 0.262 | 0.899 |
| Natural | 48.4 | 50.4 | 46.7 | 51.1 | 49.7 | 48.1 |
| − std | 0.899 | 0.141 | 0.566 | 1.14 | 0.624 | 0.0943 |

Table 30: Accuracy per prompt template for OPT-350M.

| Template | k = 0 | k = 1 | k = 5 | k = 10 | k = 15 | k = 30 |
|----------|-------|-------|-------|--------|--------|--------|
| 1 | 53.3 | 53.8 | 51.5 | 56.5 | 54.2 | 54.7 |
| 2 | 60.5 | 50.3 | 50.8 | 56.5 | 55.2 | 54.0 |
| 3 | 53.3 | 56.3 | 52.8 | 58.7 | 55.0 | 56.2 |
| 4 | 53.7 | 56.3 | 52.0 | 55.2 | 55.2 | 56.3 |
| 5 | 62.3 | 50.3 | 50.8 | 57.0 | 56.5 | 53.5 |
| 6 | 59.7 | 50.3 | 50.8 | 56.5 | 56.5 | 53.0 |
| Mean | 57.1 | 52.9 | 51.4 | 56.7 | 55.4 | 54.6 |
| − std | 3.78 | 2.71 | 0.752 | 1.04 | 0.826 | 1.26 |
| Structured | 53.4 | 55.5 | 52.1 | 56.8 | 54.8 | 55.7 |
| − std | 0.189 | 1.18 | 0.535 | 1.44 | 0.432 | 0.732 |
| Natural | 60.8 | 50.3 | 50.8 | 56.7 | 56.1 | 53.5 |
| − std | 1.09 | 7.11e-15 | 7.11e-15 | 0.236 | 0.613 | 0.408 |

Table 31: Accuracy per prompt template for OPT-1.3B.

| Template | k = 0 | k = 1 | k = 5 | k = 10 | k = 15 | k = 30 |
|----------|-------|-------|-------|--------|--------|--------|
| 1 | 57.8 | 56.2 | 55.5 | 60.2 | 59.8 | 62.7 |
| 2 | 62.2 | 57.0 | 61.2 | 61.8 | 64.8 | 67.2 |
| 3 | 60.8 | 59.5 | 57.2 | 59.7 | 60.3 | 58.2 |
| 4 | 54.8 | 55.8 | 59.2 | 56.5 | 57.0 | 54.7 |
| 5 | 62.5 | 56.2 | 59.3 | 61.7 | 65.0 | 64.5 |
| 6 | 64.0 | 53.2 | 55.8 | 59.7 | 62.7 | 62.8 |
| Mean | 60.4 | 56.3 | 58.0 | 59.9 | 61.6 | 61.7 |
| − std | 3.13 | 1.85 | 2.05 | 1.76 | 2.86 | 4.11 |
| Structured | 57.8 | 57.2 | 57.3 | 58.8 | 59.0 | 58.5 |
| − std | 2.45 | 1.66 | 1.51 | 1.64 | 1.45 | 3.27 |
| Natural | 62.9 | 55.5 | 58.8 | 61.1 | 64.2 | 64.8 |
| − std | 0.787 | 1.64 | 2.24 | 0.967 | 1.04 | 1.81 |

Table 32: Accuracy per prompt template for OPT-2.7B.

| Template | k = 0 | k = 1 | k = 5 | k = 10 | k = 15 | k = 30 |
|---|---|---|---|---|---|---|
| 1 | 54.7 | 53.0 | 53.2 | 53.8 | 54.3 | 53.7 |
| 2 | 64.0 | 60.3 | 60.2 | 60.3 | 61.3 | 64.5 |
| 3 | 55.8 | 53.3 | 55.2 | 55.8 | 57.0 | 56.5 |
| 4 | 54.5 | 53.3 | 54.8 | 55.5 | 56.8 | 57.0 |
| 5 | 64.8 | 60.7 | 60.7 | 62.2 | 64.3 | 64.3 |
| 6 | 63.5 | 60.3 | 60.0 | 60.5 | 63.3 | 63.2 |
| Mean | 59.6 | 56.8 | 57.4 | 58.0 | 59.5 | 59.9 |
| − std | 4.58 | 3.62 | 3.02 | 3.11 | 3.68 | 4.28 |
| Structured | 55.0 | 53.2 | 54.4 | 55.0 | 56.0 | 55.7 |
| − std | 0.572 | 0.141 | 0.864 | 0.881 | 1.23 | 1.45 |
| Natural | 64.1 | 60.4 | 60.3 | 61.0 | 63.0 | 64.0 |
| − std | 0.535 | 0.189 | 0.294 | 0.852 | 1.25 | 0.572 |

Table 33: Accuracy per prompt template for OPT-6.7B.

| Template | k = 0 | k = 1 | k = 5 | k = 10 | k = 15 | k = 30 |
|---|---|---|---|---|---|---|
| 1 | 55.7 | 54.3 | 60.8 | 61.2 | 61.2 | 58.5 |
| 2 | 64.2 | 68.0 | 66.8 | 65.7 | 66.3 | 66.3 |
| 3 | 54.2 | 53.5 | 59.5 | 61.2 | 63.3 | 60.5 |
| 4 | 58.8 | 56.3 | 61.8 | 62.2 | 63.5 | 63.2 |
| 5 | 64.2 | 65.2 | 66.0 | 65.2 | 67.7 | 67.5 |
| 6 | 65.0 | 63.2 | 64.8 | 64.3 | 66.3 | 65.7 |
| Mean | 60.4 | 60.1 | 63.3 | 63.3 | 64.7 | 63.6 |
| − std | 4.34 | 5.62 | 2.73 | 1.84 | 2.23 | 3.23 |
| Structured | 56.2 | 54.7 | 60.7 | 61.5 | 62.7 | 60.7 |
| − std | 1.92 | 1.18 | 0.942 | 0.471 | 1.04 | 1.93 |
| Natural | 64.5 | 65.5 | 65.9 | 65.1 | 66.8 | 66.5 |
| − std | 0.377 | 1.97 | 0.822 | 0.579 | 0.66 | 0.748 |

Table 34: Accuracy per prompt template for OPT-13B.

| Template | k = 0 | k = 1 | k = 5 | k = 10 | k = 15 | k = 30 |
|---|---|---|---|---|---|---|
| 1 | 54.7 | 64.0 | 69.8 | 68.2 | 67.8 | 62.2 |
| 2 | 68.2 | 57.8 | 69.5 | 68.0 | 66.8 | 63.7 |
| 3 | 54.3 | 62.2 | 65.2 | 63.2 | 64.3 | 66.3 |
| 4 | 58.3 | 63.3 | 64.3 | 63.7 | 63.5 | 64.0 |
| 5 | 66.0 | 58.5 | 67.2 | 65.3 | 63.7 | 62.7 |
| 6 | 64.7 | 57.5 | 68.3 | 66.2 | 64.8 | 61.5 |
| Mean | 61.0 | 60.6 | 67.4 | 65.8 | 65.1 | 63.4 |
| − std | 5.51 | 2.68 | 2.06 | 1.92 | 1.6 | 1.55 |
| Structured | 55.8 | 63.2 | 66.4 | 65.0 | 65.2 | 64.2 |
| − std | 1.8 | 0.741 | 2.41 | 2.25 | 1.87 | 1.68 |
| Natural | 66.3 | 57.9 | 68.3 | 66.5 | 65.1 | 62.6 |
| − std | 1.44 | 0.419 | 0.939 | 1.12 | 1.28 | 0.899 |

Table 35: Accuracy per prompt template for OPT-30B.

| Template | k = 0 | k = 1 | k = 5 | k = 10 | k = 15 | k = 30 |
|---|---|---|---|---|---|---|
| 1 | 62.2 | 62.7 | 66.0 | 65.2 | 65.5 | 65.0 |
| 2 | 62.0 | 58.7 | 69.0 | 65.7 | 66.3 | 69.0 |
| 3 | 60.3 | 63.5 | 62.7 | 60.8 | 60.5 | 61.5 |
| 4 | 65.0 | 66.8 | 57.8 | 57.2 | 57.2 | 56.2 |
| 5 | 60.3 | 55.8 | 70.0 | 66.0 | 67.2 | 71.0 |
| 6 | 59.0 | 54.5 | 68.3 | 65.3 | 67.7 | 70.2 |
| Mean | 61.5 | 60.3 | 65.6 | 63.4 | 64.1 | 65.5 |
| − std | 1.92 | 4.37 | 4.24 | 3.27 | 3.87 | 5.28 |
| Structured | 62.5 | 64.3 | 62.2 | 61.1 | 61.1 | 60.9 |
| − std | 1.93 | 1.77 | 3.37 | 3.27 | 3.41 | 3.62 |
| Natural | 60.4 | 56.3 | 69.1 | 65.7 | 67.1 | 70.1 |
| − std | 1.23 | 1.76 | 0.698 | 0.287 | 0.579 | 0.822 |

Table 36: Accuracy per prompt template for OPT-66B.

| Template | k = 0 | k = 1 | k = 5 | k = 10 | k = 15 | k = 30 |
|---|---|---|---|---|---|---|
| 1 | 59.3 | 56.2 | 56.7 | 56.5 | 55.7 | 54.3 |
| 2 | 66.5 | 67.3 | 65.3 | 64.2 | 67.2 | 65.2 |
| 3 | 56.5 | 64.3 | 55.5 | 55.0 | 56.2 | 52.2 |
| 4 | 62.0 | 61.5 | 66.5 | 63.0 | 61.7 | 63.7 |
| 5 | 62.5 | 66.0 | 64.8 | 63.7 | 65.7 | 65.0 |
| 6 | 61.2 | 63.8 | 60.2 | 62.5 | 64.7 | 64.7 |
| Mean | 61.3 | 63.2 | 61.5 | 60.8 | 61.9 | 60.8 |
| − std | 3.06 | 3.61 | 4.3 | 3.65 | 4.5 | 5.43 |
| Structured | 59.3 | 60.7 | 59.6 | 58.2 | 57.9 | 56.7 |
| − std | 2.25 | 3.36 | 4.93 | 3.47 | 2.72 | 5.0 |
| Natural | 63.4 | 65.7 | 63.4 | 63.5 | 65.9 | 65.0 |
| − std | 2.26 | 1.44 | 2.3 | 0.713 | 1.03 | 0.205 |

Table 37: Accuracy per prompt template for OPT-175B.

| Template | k = 0 | k = 1 | k = 5 | k = 10 | k = 15 | k = 30 |
|---|---|---|---|---|---|---|
| 1 | 56.7 | 58.0 | 64.8 | 61.0 | 65.0 | 62.3 |
| 2 | 52.7 | 53.3 | 67.3 | 63.2 | 68.0 | 65.8 |
| 3 | 54.5 | 68.5 | 60.0 | 55.3 | 57.8 | 56.7 |
| 4 | 64.0 | 66.7 | 61.5 | 58.0 | 62.0 | 58.7 |
| 5 | 52.0 | 52.0 | 65.0 | 63.8 | 67.8 | 65.2 |
| 6 | 52.2 | 51.7 | 64.7 | 63.2 | 68.0 | 66.0 |
| Mean | 55.3 | 58.4 | 63.9 | 60.8 | 64.8 | 62.4 |
| − std | 4.19 | 6.87 | 2.42 | 3.13 | 3.79 | 3.62 |
| Structured | 58.4 | 64.4 | 62.1 | 58.1 | 61.6 | 59.2 |
| − std | 4.06 | 4.58 | 2.0 | 2.33 | 2.95 | 2.32 |
| Natural | 52.3 | 52.3 | 65.7 | 63.4 | 67.9 | 65.7 |
| − std | 0.294 | 0.694 | 1.16 | 0.283 | 0.0943 | 0.34 |

Table 38: Accuracy per prompt template for Cohere-409.3M (Cohere-small).

| Template | k = 0 | k = 1 | k = 5 | k = 10 | k = 15 | k = 30 |
|---|---|---|---|---|---|---|
| 1 | 54.2 | 49.7 | 52.7 | 51.7 | 53.5 | 56.0 |
| 2 | 47.5 | 50.7 | 52.7 | 53.2 | 55.8 | 57.8 |
| 3 | 57.2 | 55.5 | 55.2 | 55.5 | 55.7 | 57.0 |
| 4 | 54.8 | 53.8 | 54.5 | 56.8 | 54.8 | 54.5 |
| 5 | 48.5 | 50.7 | 52.8 | 52.7 | 56.0 | 58.8 |
| 6 | 47.5 | 51.0 | 52.5 | 53.7 | 55.3 | 58.8 |
| Mean | 51.6 | 51.9 | 53.4 | 53.9 | 55.2 | 57.2 |
| − std | 3.91 | 2.05 | 1.05 | 1.72 | 0.847 | 1.54 |
| Structured | 55.4 | 53.0 | 54.1 | 54.7 | 54.7 | 55.8 |
| − std | 1.3 | 2.43 | 1.05 | 2.16 | 0.903 | 1.03 |
| Natural | 47.8 | 50.8 | 52.7 | 53.2 | 55.7 | 58.5 |
| − std | 0.471 | 0.141 | 0.125 | 0.408 | 0.294 | 0.471 |

Table 39: Accuracy per prompt template for Cohere-6.067B (Cohere-medium).

| Template | k = 0 | k = 1 | k = 5 | k = 10 | k = 15 | k = 30 |
|---|---|---|---|---|---|---|
| 1 | 54.7 | 54.2 | 55.3 | 51.8 | 56.3 | 55.3 |
| 2 | 61.8 | 62.8 | 64.3 | 63.8 | 65.2 | 64.7 |
| 3 | 57.2 | 53.3 | 58.5 | 55.3 | 57.8 | 55.3 |
| 4 | 56.0 | 53.3 | 57.0 | 53.2 | 55.8 | 56.7 |
| 5 | 57.8 | 60.7 | 64.0 | 64.2 | 64.7 | 64.2 |
| 6 | 56.2 | 62.8 | 66.2 | 64.0 | 62.8 | 66.0 |
| Mean | 57.3 | 57.9 | 60.9 | 58.7 | 60.4 | 60.4 |
| − std | 2.24 | 4.32 | 4.11 | 5.38 | 3.92 | 4.65 |
| Structured | 56.0 | 53.6 | 56.9 | 53.4 | 56.6 | 55.8 |
| − std | 1.02 | 0.424 | 1.31 | 1.44 | 0.85 | 0.66 |
| Natural | 58.6 | 62.1 | 64.8 | 64.0 | 64.2 | 65.0 |
| − std | 2.36 | 0.99 | 0.974 | 0.163 | 1.03 | 0.759 |

Table 40: Accuracy per prompt template for Cohere-13.12B (Cohere-large).

| Template | k = 0 | k = 1 | k = 5 | k = 10 | k = 15 | k = 30 |
|---|---|---|---|---|---|---|
| 1 | 55.3 | 57.3 | 56.3 | 55.0 | 58.5 | 59.0 |
| 2 | 59.2 | 64.2 | 68.0 | 66.3 | 64.7 | 69.5 |
| 3 | 57.2 | 62.8 | 61.0 | 59.0 | 64.2 | 62.3 |
| 4 | 55.5 | 61.3 | 56.3 | 54.0 | 59.0 | 59.8 |
| 5 | 56.8 | 64.3 | 66.7 | 64.2 | 65.7 | 69.8 |
| 6 | 59.2 | 60.7 | 66.5 | 63.7 | 65.0 | 68.3 |
| Mean | 57.2 | 61.8 | 62.5 | 60.4 | 62.9 | 64.8 |
| − std | 1.56 | 2.41 | 4.88 | 4.69 | 2.94 | 4.55 |
| Structured | 56.0 | 60.5 | 57.9 | 56.0 | 60.6 | 60.4 |
| − std | 0.852 | 2.32 | 2.22 | 2.16 | 2.58 | 1.41 |
| Natural | 58.4 | 63.1 | 67.1 | 64.7 | 65.1 | 69.2 |
| − std | 1.13 | 1.67 | 0.665 | 1.13 | 0.419 | 0.648 |

Table 41: Accuracy per prompt template for Cohere-52B (Cohere-xl).

| Template | k = 0 | k = 1 | k = 5 | k = 10 | k = 15 | k = 30 |
|---|---|---|---|---|---|---|
| 1 | 56.0 | 60.7 | 70.3 | 65.3 | 66.3 | 68.7 |
| 2 | 62.8 | 65.0 | 64.3 | 64.2 | 65.0 | 64.3 |
| 3 | 54.0 | 65.3 | 62.8 | 60.2 | 64.0 | 63.5 |
| 4 | 53.8 | 55.5 | 61.8 | 64.8 | 64.3 | 64.7 |
| 5 | 62.2 | 65.7 | 67.3 | 63.0 | 63.7 | 65.3 |
| 6 | 62.2 | 65.7 | 64.2 | 62.3 | 65.0 | 67.8 |
| Mean | 58.5 | 63.0 | 65.1 | 63.3 | 64.7 | 65.7 |
| − std | 3.97 | 3.77 | 2.87 | 1.72 | 0.855 | 1.89 |
| Structured | 54.6 | 60.5 | 65.0 | 63.4 | 64.9 | 65.6 |
| − std | 0.993 | 4.0 | 3.79 | 2.3 | 1.02 | 2.22 |
| Natural | 62.4 | 65.5 | 65.3 | 63.2 | 64.6 | 65.8 |
| − std | 0.283 | 0.33 | 1.44 | 0.785 | 0.613 | 1.47 |

Table 42: Accuracy per prompt template for GPT-3-350M (ada).

| Template | k = 0 | k = 1 | k = 5 | k = 10 | k = 15 | k = 30 |
|---|---|---|---|---|---|---|
| 1 | 55.3 | 57.2 | 58.3 | 57.5 | 58.2 | 60.5 |
| 2 | 46.7 | 56.8 | 56.3 | 59.5 | 59.2 | 61.7 |
| 3 | 54.0 | 54.5 | 53.3 | 54.0 | 56.5 | 56.7 |
| 4 | 53.5 | 52.8 | 54.7 | 56.7 | 58.8 | 59.7 |
| 5 | 49.8 | 57.3 | 55.3 | 58.5 | 58.8 | 61.8 |
| 6 | 49.5 | 57.2 | 56.3 | 60.2 | 61.5 | 61.2 |
| Mean | 51.5 | 56.0 | 55.7 | 57.7 | 58.8 | 60.3 |
| − std | 3.02 | 1.72 | 1.55 | 2.04 | 1.48 | 1.75 |
| Structured | 54.3 | 54.8 | 55.4 | 56.1 | 57.8 | 59.0 |
| − std | 0.759 | 1.81 | 2.11 | 1.5 | 0.974 | 1.64 |
| Natural | 48.7 | 57.1 | 56.0 | 59.4 | 59.8 | 61.6 |
| − std | 1.4 | 0.216 | 0.471 | 0.698 | 1.19 | 0.262 |

Table 43: Accuracy per prompt template for GPT-3-1.3B (babbage).

| Template | k = 0 | k = 1 | k = 5 | k = 10 | k = 15 | k = 30 |
|---|---|---|---|---|---|---|
| 1 | 55.7 | 60.7 | 61.0 | 59.0 | 60.7 | 57.8 |
| 2 | 63.0 | 62.5 | 65.7 | 61.7 | 63.0 | 59.3 |
| 3 | 56.2 | 59.0 | 60.5 | 59.3 | 64.8 | 61.0 |
| 4 | 53.3 | 59.7 | 60.7 | 62.5 | 65.0 | 66.7 |
| 5 | 59.2 | 62.5 | 63.7 | 61.8 | 61.5 | 58.7 |
| 6 | 59.0 | 60.2 | 64.3 | 61.2 | 62.2 | 57.7 |
| Mean | 57.7 | 60.8 | 62.6 | 60.9 | 62.9 | 60.2 |
| − std | 3.1 | 1.33 | 2.01 | 1.31 | 1.6 | 3.11 |
| Structured | 55.1 | 59.8 | 60.7 | 60.3 | 63.5 | 61.8 |
| − std | 1.27 | 0.698 | 0.205 | 1.58 | 1.98 | 3.68 |
| Natural | 60.4 | 61.7 | 64.6 | 61.6 | 62.2 | 58.6 |
| − std | 1.84 | 1.08 | 0.838 | 0.262 | 0.613 | 0.66 |

Table 44: Accuracy per prompt template for GPT-3-6.7B (curie).

| Template | k = 0 | k = 1 | k = 5 | k = 10 | k = 15 | k = 30 |
|---|---|---|---|---|---|---|
| 1 | 53.3 | 58.3 | 63.0 | 64.8 | 67.7 | 64.0 |
| 2 | 57.5 | 65.2 | 63.2 | 65.3 | 65.8 | 65.2 |
| 3 | 57.0 | 54.2 | 59.2 | 61.2 | 60.8 | 59.3 |
| 4 | 53.3 | 61.7 | 62.8 | 63.8 | 64.7 | 60.7 |
| 5 | 55.3 | 64.2 | 62.5 | 64.5 | 65.8 | 63.7 |
| 6 | 52.5 | 63.5 | 63.7 | 64.0 | 66.2 | 64.3 |
| Mean | 54.8 | 61.2 | 62.4 | 63.9 | 65.2 | 62.9 |
| − std | 1.92 | 3.83 | 1.48 | 1.32 | 2.14 | 2.12 |
| Structured | 54.5 | 58.1 | 61.7 | 63.3 | 64.4 | 61.3 |
| − std | 1.74 | 3.07 | 1.75 | 1.52 | 2.82 | 1.97 |
| Natural | 55.1 | 64.3 | 63.1 | 64.6 | 65.9 | 64.4 |
| − std | 2.05 | 0.698 | 0.492 | 0.535 | 0.189 | 0.616 |

Table 45: Accuracy per prompt template for GPT-3-175B (davinci).

| Template | k = 0 | k = 1 | k = 5 | k = 10 | k = 15 | k = 30 |
|---|---|---|---|---|---|---|
| 1 | 61.2 | 67.3 | 66.3 | 62.7 | 66.7 | 66.2 |
| 2 | 53.7 | 65.3 | 68.8 | 69.3 | 71.0 | 69.7 |
| 3 | 58.7 | 65.8 | 68.2 | 64.7 | 65.0 | 65.3 |
| 4 | 64.0 | 62.8 | 71.3 | 68.7 | 66.2 | 67.8 |
| 5 | 54.2 | 66.3 | 69.0 | 70.0 | 70.0 | 70.8 |
| 6 | 51.7 | 66.7 | 68.7 | 68.3 | 71.0 | 70.0 |
| Mean | 57.2 | 65.7 | 68.7 | 67.3 | 68.3 | 68.3 |
| − std | 4.4 | 1.44 | 1.46 | 2.65 | 2.43 | 2.03 |
| Structured | 61.3 | 65.3 | 68.6 | 65.4 | 66.0 | 66.4 |
| − std | 2.16 | 1.87 | 2.06 | 2.49 | 0.713 | 1.03 |
| Natural | 53.2 | 66.1 | 68.8 | 69.2 | 70.7 | 70.2 |
| − std | 1.08 | 0.589 | 0.125 | 0.698 | 0.471 | 0.464 |

Table 46: Accuracy per prompt template for BlenderBot-90M.

| Template | k = 0 | k = 1 | k = 5 | k = 10 | k = 15 | k = 30 |
|---|---|---|---|---|---|---|
| 1 | 46.7 | 51.5 | 46.7 | 46.7 | 46.5 | 46.5 |
| 2 | 46.7 | 51.3 | 46.5 | 46.7 | 46.7 | 46.7 |
| 3 | 46.7 | 46.7 | 46.7 | 46.7 | 46.3 | 46.8 |
| 4 | 46.7 | 46.7 | 46.7 | 46.7 | 46.5 | 46.7 |
| 5 | 46.7 | 50.0 | 46.7 | 46.7 | 46.7 | 46.7 |
| 6 | 46.5 | 53.5 | 46.3 | 46.7 | 46.7 | 46.7 |
| Mean | 46.7 | 49.9 | 46.6 | 46.7 | 46.6 | 46.7 |
| − std | 0.0745 | 2.52 | 0.153 | 7.11e-15 | 0.149 | 0.0898 |
| Structured | 46.7 | 48.3 | 46.7 | 46.7 | 46.4 | 46.7 |
| − std | 7.11e-15 | 2.26 | 7.11e-15 | 7.11e-15 | 0.0943 | 0.125 |
| Natural | 46.6 | 51.6 | 46.5 | 46.7 | 46.7 | 46.7 |
| − std | 0.0943 | 1.44 | 0.163 | 7.11e-15 | 7.11e-15 | 7.11e-15 |

Table 47: Accuracy per prompt template for BlenderBot-2.7B.

| Template | k = 0 | k = 1 | k = 5 | k = 10 | k = 15 | k = 30 |
|---|---|---|---|---|---|---|
| 1 | 54.0 | 53.2 | 53.3 | 53.0 | 52.8 | 53.3 |
| 2 | 53.3 | 53.3 | 53.3 | 53.3 | 53.3 | 53.3 |
| 3 | 53.2 | 53.2 | 53.3 | 53.2 | 53.2 | 53.2 |
| 4 | 53.5 | 53.5 | 53.5 | 53.3 | 52.8 | 53.0 |
| 5 | 53.3 | 53.3 | 53.3 | 53.3 | 53.3 | 53.3 |
| 6 | 53.3 | 53.3 | 53.3 | 53.3 | 53.3 | 53.3 |
| Mean | 53.4 | 53.3 | 53.3 | 53.2 | 53.1 | 53.2 |
| − std | 0.269 | 0.1 | 0.0745 | 0.111 | 0.227 | 0.111 |
| Structured | 53.6 | 53.3 | 53.4 | 53.2 | 52.9 | 53.2 |
| − std | 0.33 | 0.141 | 0.0943 | 0.125 | 0.189 | 0.125 |
| Natural | 53.3 | 53.3 | 53.3 | 53.3 | 53.3 | 53.3 |
| − std | 7.11e-15 | 7.11e-15 | 7.11e-15 | 7.11e-15 | 7.11e-15 | 7.11e-15 |

Table 48: Accuracy per prompt template for BlenderBot-9.4B.

| Template | k = 0 | k = 1 | k = 5 | k = 10 | k = 15 | k = 30 |
|---|---|---|---|---|---|---|
| 1 | 53.7 | 51.5 | 53.0 | 53.0 | 53.0 | 54.0 |
| 2 | 53.2 | 53.8 | 54.2 | 52.5 | 52.2 | 52.2 |
| 3 | 53.3 | 49.7 | 52.0 | 54.0 | 54.2 | 55.5 |
| 4 | 54.0 | 55.3 | 52.5 | 54.0 | 53.5 | 53.7 |
| 5 | 53.3 | 52.8 | 53.5 | 53.2 | 53.5 | 53.3 |
| 6 | 52.7 | 52.0 | 51.7 | 53.5 | 52.8 | 53.7 |
| Mean | 53.4 | 52.5 | 52.8 | 53.4 | 53.2 | 53.7 |
| − std | 0.407 | 1.77 | 0.859 | 0.537 | 0.63 | 0.978 |
| Structured | 53.7 | 52.2 | 52.5 | 53.7 | 53.6 | 54.4 |
| − std | 0.287 | 2.33 | 0.408 | 0.471 | 0.492 | 0.787 |
| Natural | 53.1 | 52.9 | 53.1 | 53.1 | 52.8 | 53.1 |
| − std | 0.262 | 0.736 | 1.05 | 0.419 | 0.531 | 0.634 |

Table 49: Accuracy per prompt template for T0-3B.

| Template | k = 0 | k = 1 | k = 5 | k = 10 | k = 15 | k = 30 |
|---|---|---|---|---|---|---|
| 1 | 48.7 | 49.5 | 46.5 | 46.7 | 46.7 | 46.7 |
| 2 | 46.7 | 47.5 | 46.7 | 46.7 | 46.7 | 46.7 |
| 3 | 49.2 | 48.3 | 46.7 | 46.7 | 46.7 | 46.7 |
| 4 | 51.7 | 49.0 | 46.7 | 46.7 | 46.7 | 46.7 |
| 5 | 46.7 | 49.2 | 46.7 | 46.7 | 46.7 | 46.7 |
| 6 | 46.7 | 49.8 | 46.8 | 46.7 | 46.7 | 46.7 |
| Mean | 48.3 | 48.9 | 46.7 | 46.7 | 46.7 | 46.7 |
| − std | 1.84 | 0.773 | 0.0898 | 7.11e-15 | 7.11e-15 | 7.11e-15 |
| Structured | 49.9 | 48.9 | 46.6 | 46.7 | 46.7 | 46.7 |
| − std | 1.31 | 0.492 | 0.0943 | 7.11e-15 | 7.11e-15 | 7.11e-15 |
| Natural | 46.7 | 48.8 | 46.7 | 46.7 | 46.7 | 46.7 |
| − std | 7.11e-15 | 0.974 | 0.0471 | 7.11e-15 | 7.11e-15 | 7.11e-15 |

Table 50: Accuracy per prompt template for T0-11B.

| Template | k = 0 | k = 1 | k = 5 | k = 10 | k = 15 | k = 30 |
|---|---|---|---|---|---|---|
| 1 | 57.5 | 47.7 | 47.3 | 46.8 | 46.7 | 46.7 |
| 2 | 49.3 | 47.5 | 46.7 | 46.7 | 46.8 | 46.7 |
| 3 | 65.3 | 48.8 | 47.3 | 46.7 | 46.7 | 46.7 |
| 4 | 63.8 | 48.0 | 47.0 | 46.7 | 46.7 | 46.7 |
| 5 | 48.0 | 47.2 | 46.7 | 46.7 | 47.0 | 46.8 |
| 6 | 49.7 | 47.5 | 47.0 | 46.8 | 47.0 | 47.0 |
| Mean | 55.6 | 47.8 | 47.0 | 46.7 | 46.8 | 46.8 |
| − std | 7.04 | 0.515 | 0.245 | 0.0471 | 0.134 | 0.111 |
| Structured | 62.2 | 48.2 | 47.2 | 46.7 | 46.7 | 46.7 |
| − std | 3.38 | 0.464 | 0.141 | 0.0471 | 7.11e-15 | 7.11e-15 |
| Natural | 49.0 | 47.4 | 46.8 | 46.7 | 46.9 | 46.8 |
| − std | 0.726 | 0.141 | 0.141 | 0.0471 | 0.0943 | 0.125 |

Table 51: Accuracy per prompt template for Flan-T5-780M.

| Template | k = 0 | k = 1 | k = 5 | k = 10 | k = 15 | k = 30 |
|---|---|---|---|---|---|---|
| 1 | 64.5 | 63.3 | 62.2 | 60.7 | 61.5 | 60.2 |
| 2 | 66.5 | 65.8 | 65.3 | 62.8 | 65.5 | 65.0 |
| 3 | 61.7 | 60.2 | 58.8 | 60.8 | 59.8 | 59.7 |
| 4 | 58.0 | 50.2 | 50.7 | 51.3 | 52.3 | 54.8 |
| 5 | 63.8 | 69.0 | 64.3 | 63.2 | 65.2 | 65.5 |
| 6 | 65.3 | 68.8 | 64.8 | 62.3 | 64.7 | 63.8 |
| Mean | 63.3 | 62.9 | 61.0 | 60.2 | 61.5 | 61.5 |
| − std | 2.79 | 6.44 | 5.1 | 4.08 | 4.61 | 3.73 |
| Structured | 61.4 | 57.9 | 57.2 | 57.6 | 57.9 | 58.2 |
| − std | 2.66 | 5.59 | 4.82 | 4.45 | 4.0 | 2.44 |
| Natural | 65.2 | 67.9 | 64.8 | 62.8 | 65.1 | 64.8 |
| − std | 1.1 | 1.46 | 0.408 | 0.368 | 0.33 | 0.713 |

Table 52: Accuracy per prompt template for Flan-T5-3B.

| Template | k = 0 | k = 1 | k = 5 | k = 10 | k = 15 | k = 30 |
|---|---|---|---|---|---|---|
| 1 | 54.7 | 58.8 | 56.8 | 56.7 | 57.5 | 60.0 |
| 2 | 51.2 | 50.8 | 59.0 | 59.2 | 59.0 | 59.7 |
| 3 | 54.8 | 51.3 | 49.7 | 49.0 | 48.7 | 48.5 |
| 4 | 55.3 | 50.0 | 48.0 | 49.0 | 49.3 | 50.8 |
| 5 | 51.0 | 54.3 | 57.2 | 58.0 | 58.0 | 57.8 |
| 6 | 48.0 | 51.2 | 58.7 | 59.0 | 58.0 | 59.8 |
| Mean | 52.5 | 52.7 | 54.9 | 55.1 | 55.1 | 56.1 |
| − std | 2.65 | 3.02 | 4.37 | 4.42 | 4.33 | 4.67 |
| Structured | 54.9 | 53.4 | 51.5 | 51.6 | 51.8 | 53.1 |
| − std | 0.262 | 3.88 | 3.81 | 3.63 | 4.01 | 4.97 |
| Natural | 50.1 | 52.1 | 58.3 | 58.7 | 58.3 | 59.1 |
| − std | 1.46 | 1.56 | 0.787 | 0.525 | 0.471 | 0.92 |

Table 53: Accuracy per prompt template for Flan-T5-11B.

| Template | k = 0 | k = 1 | k = 5 | k = 10 | k = 15 | k = 30 |
|---|---|---|---|---|---|---|
| 1 | 64.3 | 61.0 | 63.7 | 65.0 | 62.5 | 64.3 |
| 2 | 61.5 | 59.7 | 63.2 | 62.3 | 64.0 | 68.0 |
| 3 | 56.5 | 63.0 | 60.2 | 57.3 | 56.7 | 56.8 |
| 4 | 61.7 | 47.7 | 51.7 | 50.3 | 50.3 | 49.5 |
| 5 | 61.5 | 55.8 | 64.8 | 64.7 | 65.5 | 66.3 |
| 6 | 59.2 | 57.5 | 66.3 | 63.7 | 66.0 | 67.7 |
| Mean | 60.8 | 57.4 | 61.7 | 60.5 | 60.8 | 62.1 |
| − std | 2.42 | 4.94 | 4.82 | 5.25 | 5.62 | 6.78 |
| Structured | 60.8 | 57.2 | 58.5 | 57.5 | 56.5 | 56.9 |
| − std | 3.24 | 6.79 | 5.04 | 6.0 | 4.98 | 6.04 |
| Natural | 60.7 | 57.7 | 64.8 | 63.6 | 65.2 | 67.3 |
| − std | 1.08 | 1.6 | 1.27 | 0.984 | 0.85 | 0.741 |

Table 54: Accuracy per prompt template for InstructGPT-3-350M (text-ada-001).

| Template | k = 0 | k = 1 | k = 5 | k = 10 | k = 15 | k = 30 |
|---|---|---|---|---|---|---|
| 1 | 60.8 | 62.8 | 60.8 | 59.0 | 58.7 | 58.8 |
| 2 | 50.7 | 56.3 | 54.8 | 56.0 | 57.7 | 52.7 |
| 3 | 63.7 | 58.5 | 60.8 | 59.0 | 56.7 | 57.5 |
| 4 | 61.8 | 56.3 | 59.3 | 58.3 | 61.0 | 56.7 |
| 5 | 53.3 | 55.5 | 55.2 | 55.7 | 58.0 | 54.3 |
| 6 | 48.7 | 54.7 | 54.7 | 56.2 | 57.7 | 53.5 |
| Mean | 56.5 | 57.3 | 57.6 | 57.4 | 58.3 | 55.6 |
| − std | 5.82 | 2.7 | 2.75 | 1.43 | 1.34 | 2.22 |
| Structured | 62.1 | 59.2 | 60.3 | 58.8 | 58.8 | 57.7 |
| − std | 1.2 | 2.7 | 0.707 | 0.33 | 1.76 | 0.865 |
| Natural | 50.9 | 55.5 | 54.9 | 56.0 | 57.8 | 53.5 |
| − std | 1.88 | 0.653 | 0.216 | 0.205 | 0.141 | 0.653 |

Table 55: Accuracy per prompt template for InstructGPT-3-1.3B (text-babbage-001).

| Template | k = 0 | k = 1 | k = 5 | k = 10 | k = 15 | k = 30 |
|---|---|---|---|---|---|---|
| 1 | 67.5 | 64.0 | 66.3 | 63.0 | 64.0 | 64.7 |
| 2 | 63.0 | 62.5 | 66.2 | 64.2 | 66.5 | 68.2 |
| 3 | 65.3 | 65.2 | 66.0 | 63.2 | 64.7 | 64.5 |
| 4 | 65.2 | 63.5 | 65.7 | 62.7 | 63.0 | 64.8 |
| 5 | 61.8 | 64.3 | 66.5 | 64.0 | 66.3 | 67.8 |
| 6 | 64.0 | 63.8 | 66.2 | 64.2 | 66.7 | 66.0 |
| Mean | 64.5 | 63.9 | 66.1 | 63.6 | 65.2 | 66.0 |
| − std | 1.82 | 0.815 | 0.25 | 0.605 | 1.4 | 1.5 |
| Structured | 66.0 | 64.2 | 66.0 | 63.0 | 63.9 | 64.7 |
| − std | 1.06 | 0.713 | 0.245 | 0.205 | 0.698 | 0.125 |
| Natural | 62.9 | 63.5 | 66.3 | 64.1 | 66.5 | 67.3 |
| − std | 0.899 | 0.759 | 0.141 | 0.0943 | 0.163 | 0.957 |

Table 56: Accuracy per prompt template for InstructGPT-3-6.7B (text-curie-001).

| Template | k = 0 | k = 1 | k = 5 | k = 10 | k = 15 | k = 30 |
|---|---|---|---|---|---|---|
| 1 | 70.7 | 70.2 | 72.5 | 70.8 | 70.8 | 70.7 |
| 2 | 66.5 | 59.3 | 70.3 | 69.7 | 68.3 | 71.2 |
| 3 | 73.2 | 70.2 | 73.5 | 69.7 | 71.8 | 69.7 |
| 4 | 71.3 | 68.0 | 71.0 | 69.8 | 71.0 | 69.0 |
| 5 | 65.5 | 58.8 | 70.0 | 70.2 | 68.5 | 70.7 |
| 6 | 66.5 | 59.8 | 70.7 | 70.8 | 69.0 | 70.8 |
| Mean | 69.0 | 64.4 | 71.3 | 70.2 | 69.9 | 70.4 |
| − std | 2.9 | 5.14 | 1.25 | 0.478 | 1.35 | 0.754 |
| Structured | 71.7 | 69.5 | 72.3 | 70.1 | 71.2 | 69.8 |
| − std | 1.07 | 1.04 | 1.03 | 0.497 | 0.432 | 0.698 |
| Natural | 66.2 | 59.3 | 70.3 | 70.2 | 68.6 | 70.9 |
| − std | 0.471 | 0.408 | 0.287 | 0.45 | 0.294 | 0.216 |

Table 57: Accuracy per prompt template for InstructGPT-3-175B (text-davinci-001).

| Template | k = 0 | k = 1 | k = 5 | k = 10 | k = 15 | k = 30 |
|---|---|---|---|---|---|---|
| 1 | 76.5 | 73.7 | 75.7 | 75.7 | 76.3 | 76.8 |
| 2 | 72.0 | 72.5 | 74.3 | 75.2 | 76.0 | 75.3 |
| 3 | 74.8 | 74.2 | 75.7 | 77.2 | 75.8 | 76.8 |
| 4 | 68.0 | 70.2 | 72.8 | 72.8 | 73.3 | 75.0 |
| 5 | 72.5 | 73.2 | 74.3 | 74.3 | 75.3 | 75.7 |
| 6 | 70.0 | 72.7 | 74.3 | 74.7 | 75.0 | 75.3 |
| Mean | 72.3 | 72.7 | 74.5 | 75.0 | 75.3 | 75.8 |
| − std | 2.82 | 1.28 | 0.991 | 1.34 | 0.986 | 0.724 |
| Structured | 73.1 | 72.7 | 74.7 | 75.2 | 75.1 | 76.2 |
| − std | 3.67 | 1.78 | 1.37 | 1.83 | 1.31 | 0.849 |
| Natural | 71.5 | 72.8 | 74.3 | 74.7 | 75.4 | 75.4 |
| − std | 1.08 | 0.294 | 0.0 | 0.368 | 0.419 | 0.189 |

Table 58: Accuracy per prompt template for text-davinci-002-unknown.

| Template | k = 0 | k = 1 | k = 5 | k = 10 | k = 15 | k = 30 |
|---|---|---|---|---|---|---|
| 1 | 73.7 | 76.2 | 80.2 | 79.5 | 79.8 | 80.7 |
| 2 | 69.5 | 73.5 | 78.2 | 78.5 | 76.7 | 79.8 |
| 3 | 73.0 | 78.7 | 82.8 | 82.8 | 82.7 | 82.8 |
| 4 | 71.3 | 79.7 | 80.5 | 80.8 | 82.0 | 81.5 |
| 5 | 67.5 | 72.5 | 79.2 | 79.2 | 77.0 | 79.8 |
| 6 | 68.5 | 73.2 | 76.5 | 76.5 | 76.2 | 79.2 |
| Mean | 70.6 | 75.6 | 79.6 | 79.5 | 79.1 | 80.6 |
| − std | 2.28 | 2.79 | 1.96 | 1.94 | 2.6 | 1.22 |
| Structured | 72.7 | 78.2 | 81.2 | 81.0 | 81.5 | 81.7 |
| − std | 1.01 | 1.47 | 1.16 | 1.36 | 1.24 | 0.865 |
| Natural | 68.5 | 73.1 | 78.0 | 78.1 | 76.6 | 79.6 |
| − std | 0.816 | 0.419 | 1.11 | 1.14 | 0.33 | 0.283 |

Table 59: Timestamp each was evaluated through OpenAI's API.

| model | timestamp |
|---|---|
| GPT-3-ada/0-shot | 2022-09-22 13:13:29 |
| GPT-3-ada/1-shot | 2022-09-22 15:11:13 |
| GPT-3-ada/5-shot | 2022-09-22 15:40:12 |
| GPT-3-ada/10-shot | 2022-09-22 18:14:18 |
| GPT-3-ada/15-shot | 2022-09-22 19:15:29 |
| GPT-3-ada/30-shot | 2022-09-22 22:47:58 |
| GPT-3-babbage/0-shot | 2022-09-22 23:19:05 |
| GPT-3-babbage/1-shot | 2022-09-22 23:39:53 |
| GPT-3-babbage/5-shot | 2022-09-23 00:01:32 |
| GPT-3-babbage/10-shot | 2022-09-23 00:24:27 |
| GPT-3-babbage/15-shot | 2022-09-23 00:49:13 |
| GPT-3-babbage/30-shot | 2022-09-23 01:15:44 |
| GPT-3-curie/0-shot | 2022-09-22 14:04:32 |
| GPT-3-curie/1-shot | 2022-09-23 02:09:14 |
| GPT-3-curie/5-shot | 2022-09-23 02:32:20 |
| GPT-3-curie/10-shot | 2022-09-23 02:56:43 |
| GPT-3-curie/15-shot | 2022-09-23 03:23:19 |
| GPT-3-curie/30-shot | 2022-09-23 03:52:30 |
| GPT-3-davinci/0-shot | 2022-09-22 12:21:48 |
| GPT-3-davinci/1-shot | 2022-09-23 14:27:15 |
| GPT-3-davinci/5-shot | 2022-09-23 15:10:40 |
| GPT-3-davinci/10-shot | 2022-09-23 16:04:53 |
| GPT-3-davinci/15-shot | 2022-09-23 17:17:04 |
| GPT-3-davinci/30-shot | 2022-09-23 18:36:38 |
| OpenAI-text-ada-001/0-shot | 2022-08-17 16:59:45 |
| OpenAI-text-ada-001/1-shot | 2022-08-17 18:23:12 |
| OpenAI-text-ada-001/5-shot | 2022-08-17 19:16:48 |
| OpenAI-text-ada-001/10-shot | 2022-08-17 20:24:16 |
| OpenAI-text-ada-001/15-shot | 2022-08-17 21:21:46 |
| OpenAI-text-ada-001/30-shot | 2022-08-17 22:44:47 |
| OpenAI-text-babbage-001/0-shot | 2022-08-17 11:50:44 |
| OpenAI-text-babbage-001/1-shot | 2022-08-17 12:22:08 |
| OpenAI-text-babbage-001/5-shot | 2022-08-17 12:50:59 |
| OpenAI-text-babbage-001/10-shot | 2022-08-17 13:27:52 |
| OpenAI-text-babbage-001/15-shot | 2022-08-17 14:57:43 |
| OpenAI-text-babbage-001/30-shot | 2022-08-17 15:45:16 |
| OpenAI-text-curie-001/0-shot | 2022-08-18 04:39:55 |
| OpenAI-text-curie-001/1-shot | 2022-08-18 05:10:17 |
| OpenAI-text-curie-001/5-shot | 2022-08-18 05:40:56 |
| OpenAI-text-curie-001/10-shot | 2022-08-18 06:15:28 |
| OpenAI-text-curie-001/15-shot | 2022-08-18 06:53:09 |
| OpenAI-text-curie-001/30-shot | 2022-08-18 07:35:40 |
| OpenAI-text-davinci-001/0-shot | 2022-08-26 20:26:21 |
| OpenAI-text-davinci-001/1-shot | 2022-08-26 21:02:31 |
| OpenAI-text-davinci-001/5-shot | 2022-08-26 21:35:19 |
| OpenAI-text-davinci-001/10-shot | 2022-08-27 07:14:02 |
| OpenAI-text-davinci-001/15-shot | 2022-08-27 07:58:25 |
| OpenAI-text-davinci-001/30-shot | 2022-08-27 08:44:42 |
| OpenAI-text-davinci-002/0-shot | 2022-08-10 21:41:50 |
| OpenAI-text-davinci-002/1-shot | 2022-08-11 10:04:17 |
| OpenAI-text-davinci-002/5-shot | 2022-08-12 15:41:45 |
| OpenAI-text-davinci-002/10-shot | 2022-08-12 16:41:14 |
| OpenAI-text-davinci-002/15-shot | 2022-08-16 12:11:43 |
| OpenAI-text-davinci-002/30-shot | 2022-08-16 14:35:38 |

Table 60: Timestamp each model was evaluated through Cohere's API.

| model | timestamp |
|---|---|
| Cohere-small/0-shot | 2022-08-16 22:22:17 |
| Cohere-small/1-shot | 2022-08-17 08:22:43 |
| Cohere-small/5-shot | 2022-08-17 09:19:57 |
| Cohere-small/10-shot | 2022-08-17 10:43:53 |
| Cohere-small/15-shot | 2022-08-17 12:53:02 |
| Cohere-small/30-shot | 2022-08-17 13:46:08 |
| Cohere-medium/0-shot | 2022-08-17 15:14:02 |
| Cohere-medium/1-shot | 2022-08-17 16:00:21 |
| Cohere-medium/5-shot | 2022-08-17 18:23:38 |
| Cohere-medium/10-shot | 2022-08-17 19:16:00 |
| Cohere-medium/15-shot | 2022-08-17 20:24:12 |
| Cohere-medium/30-shot | 2022-08-17 21:20:28 |
| Cohere-large/0-shot | 2022-08-17 22:47:49 |
| Cohere-large/1-shot | 2022-08-17 23:27:00 |
| Cohere-large/5-shot | 2022-08-18 00:10:08 |
| Cohere-large/10-shot | 2022-08-18 00:56:55 |
| Cohere-large/15-shot | 2022-08-18 01:48:30 |
| Cohere-large/30-shot | 2022-08-18 02:47:14 |
| Cohere-xl/0-shot | 2022-07-29 |
| Cohere-xl/1-shot | 2022-07-31 |
| Cohere-xl/5-shot | 2022-08-02 |
| Cohere-xl/10-shot | 2022-08-02 15:16:45 |
| Cohere-xl/15-shot | 2022-08-07 13:55:44 |
| Cohere-xl/30-shot | 2022-08-16 19:51:08 |

