# OpenReview forum: "Large language models are not zero-shot communicators"
_ICLR.cc/2023/Conference — Submitted to ICLR 2023_

### Official Review · Reviewer_kGh3 · 2022-10-24

**Confidence:** 4
**Correctness:** 3
**Technical Novelty And Significance:** 3
**Empirical Novelty And Significance:** 3
**Recommendation:** 5

**Clarity, Quality, Novelty And Reproducibility:**

The idea is clearly presented, and the paper is easy to read. The appendix section provides more details about the evaluation results. The result seems reproduceable without much effort.

**Strength And Weaknesses:**

Strength:
The paper presents an interesting result of pretrained LMs performance on the binary implicature resolution task. While we are seeing promising performance with the large LMs, especially with the newest ones like OpenAI's, there is still a gap with human performance on this implicature resolution task.

Weaknesses:
The claim that large LMs are not zero-shot communicators is a little strong because it is only based on a single dataset. There will be questions regarding whether this dataset is representative? the dataset size (600 samples) is enough or not? and even the quality of the samples/labels?

For the samples that require context understanding ("Particularised"), authors can show more case studies and analyze the predictions on these samples. Also, I think it is better to plot the absolute accuracy scores in Figure 3 to illustrate whether the performance on this type of samples will increase with the model size.

The claim that "increasing the model size and prompt size is unlikely to close the gap" seems not true for the recent model from OpenAI. Evidence in Figures 2 and 4 show that we are seeing performance gain when going to bigger model or having more few-shot examples in the prompt.

Furthermore, the current paper does not have enough analysis to understand why the current LM models fail and hence suggest directions for improvement.

**Summary Of The Paper:**

Authors evaluate pretrained large LMs on the binary implicature resolution. The results reveal that there is still a big gap between human performance and LM's on this task, indicating the challenge to interpret the language in context of the LMs. They also show that increasing the model size and prompt size (through few-shot prompting) is unlikely to close the gap, especially for the samples that require more context understanding.

**Summary Of The Review:**

Although this work reveals an interesting result when evaluating LMs for conversation context understanding, this evaluation is on a single test set, hence it is not strong enough to make a claim on the general effectiveness LMs. It is also expected to see more analysis to understand why the current LMs model fail on this test and hence have some potential directions for improvements.

---

> ### Author Response · Authors · 2022-11-10
> **(Part 2/2) Response to reviewer kGh3 -- Points 3 & 4**
>
> ### Point 3 -- on the effect of scale
> *Point 3*: *“The claim that "increasing the model size and prompt size is unlikely to close the gap" seems not true for the recent model from OpenAI. Evidence in Figures 2 and 4 show that we are seeing performance gain when going to bigger model or having more few-shot examples in the prompt.”* Thanks for pointing this out! We agree that we do not have ironclad evidence for this claim. What we mean is that it seems unlikely that the zero-shot gap will be closed by model size from Figure 2 (the model scaling plot plateaus around 70% accuracy, and an extrapolation analysis we did by fitting a curve to 0-shot InstructGPT-3 tells us it might reach 86% accuracy around 70 trillion parameters with this trend), and none of the models improve significantly from $k>5$. However, to nuance this claim in our work we have added the following sentence to the conclusion in the uploaded revision *“Additionally, to substantiate the hypothesis that model size will not close the gap future work with larger model sizes is needed.”*.
>
> ### Point 4 -- not enough analysis to understand why LMs fail
> *Point 4*: *“Furthermore, the current paper does not have enough analysis to understand why the current LM models fail and hence suggest directions for improvement.”*. While this is true, we respectfully disagree with the implication that this is a detracting factor for our paper or that this is something that we can be expected to do. We hope to clarify this view below.
>
> Firstly, doing *“analysis to understand why the current LM models fail and hence suggest directions for improvement”* is simply not something we are capable of engaging in. For several of the LLMs studied in this paper, we do not have direct access to the models and cannot even confirm that the input string to the model was exactly what we submitted through the API. Even when we do have access to the weights, only three of the nine model families we studied (flan-T5, T0, and EleutherAI) are trained on publicly reproducible data. Moreover, to understand why these models fail, a proper analysis with ablations of the architecture and training-method is necessary, and this is computationally intractable for most of the NLP research community.
>
> Secondly, we believe that our work is interesting as a separate line of inquiry; pointing out the failure modes of large models, and showing where they need to improve in order to work towards the collective goal of human language understanding by models. We do a comprehensive set of experiments on a large set of models that allow investigating the effect of model scaling and in-context prompting on four different groups of LLMs, and show a significant gap with human performance on a task that is ubiquitously part of our day-to-day conversations. A large part of our efforts have been put into manually categorising the test set and analysing on what type of examples the models fail. This helped us show that the performance increase the models do see from model size is driven by the simplest examples in the dataset, hence future work should focus on more complex implicature benchmarks.
>
> ### Concluding remarks
>
> We hope that the reviewer’s points of weakness are adequately addressed by the title change, the analysis with the absolute type accuracy showing accuracy on Particularised examples does not increase with model size, as well as the nuance added to the revision for the claim that model scale will not help in the future. If not, we would be grateful if you could indicate outstanding points of concern which stand in the way of you raising your score, and we will gladly seek to address them.

---

> ### Author Response · Authors · 2022-11-10
> **Part (1/2) Response to reviewer kGh3 -- Points 1 & 2**
>
> We thank **reviewer kGh3** for taking the time to read and comment on our work, saying that  *“the paper presents an interesting result”*, the technical and empirical contributions are significant, and *“The idea is clearly presented, and the paper is easy to read”*. We hope to address your constructive feedback below, and discuss how this has helped us improve the paper in response (a full revision has been uploaded as well as a pdf with the revisions highlighted in red in the supplementary material folder).
>
> ### Point 1 -- Claim that LLMs are not zero-shot communicators & use of 1 test set with 600 examples
> *Point 1*: *“The claim that large LMs are not zero-shot communicators is a little strong because it is only based on a single dataset. The dataset size (600 samples) is enough or not? and even the quality of the samples/labels?”* For the claim that large LMs are not zero-shot communicators, we refer the reviewer to the [common response to all reviewers above](https://openreview.net/forum?id=WgbcOQMNXB&noteId=UcFHII1IXi). We explain our rationale for this claim, but accept a change to the title is warranted and suggest an alternative title not containing this claim that we would love to get your feedback on.
>
> Regarding the quality of the samples and labels we’d like to point out that the dataset has been curated from TOEFL English proficiency tests and 45 movie scripts. The labels are manually added by George and Mamidi (2020), who note that they opted to manually label to get a better quality dataset than through crowdsourcing. Additionally, the high inter-annotator agreement in our own study points to humans agreeing on the labels of a large portion of the data in this benchmark.
>
> We believe 600 test examples are representative enough to say that LLMs struggle with binary implicature resolution compared to humans. The high inter-annotator agreement of humans coupled with the fact that the st. dev. among human accuracy is only 2.3%, shows that the gap of 14% accuracy between humans and models is a significant one. Additionally, the task we are looking at is a binary task, meaning we have roughly ~300 test examples for each label. Finally, we agree that the test set will not cover the full natural distribution of implicatures found in conversation, but failing on this implicature test set involves failing a necessary but not sufficient set of conditions for asserting that LLMs do understand implicature. In the manuscript’s conclusion we mention the future work requirements to further benchmark the breadth of implicature understanding appearing in natural conversation.
>
> ### Point 2 -- Absolute accuracy
> *Point 2*: *“For the samples that require context understanding ("Particularised"), authors can show more case studies and analyze the predictions on these samples. Also, I think it is better to plot the absolute accuracy scores in Figure 3 to illustrate whether the performance on this type of samples will increase with the model size.”*. Thanks for pointing this out! We have made an absolute plot of Figure 3, which can be found in in appendix F.4 of the uploaded revision (Figure 7). What we can see from this plot is that indeed for InstructGPT-3 the performance on the Particularised label decreases with model size, and for Cohere the performance on the Particularised label stays the same with model size (which is random performance, so it won’t decrease further).
>
> With regards the request for “more case studies and analyze the predictions on these samples”. The type label analysis shows that the performance increase with model size is driven by simple examples in the dataset (Generalised implicatures), which is done by manually labeling the test set and analysing where the models struggle. We would be grateful if the reviewer can clarify what kind of additional case studies and prediction analyses they mean.

---

### Official Review · Reviewer_J3vy · 2022-10-24

**Confidence:** 4
**Correctness:** 3
**Technical Novelty And Significance:** 3
**Empirical Novelty And Significance:** 3
**Recommendation:** 8

**Clarity, Quality, Novelty And Reproducibility:**

This task and experiment design in this paper is very straightforward, and it is presented clearly. Given access to the models used in this paper, I think the results should be reproducible.

**Strength And Weaknesses:**

# Strength:

1. It is interesting to understand how LLM understand implicatures which is acquired through social learning for humans. The contrastive task design is neat and effective.
2. The experiments make sense and the results are informative.

# Weakness

1. Given the small size of the dataset, some of the results for smaller categories may not be significant. It would be nice if the authors could calculate the uncertainty for Figure 3 & 4.
2. The title is too eye-catching-oriented and a little disconnected to the main claim of this paper. I would suggest use "Measuring implicature understanding" or similar ones to more clearly state the purpose of this paper. Communication is a much broader concepts than implicature. Not being able to understand implicature than average humans doesn't mean LLMs cannot communicate.

**Summary Of The Paper:**

In this paper, the authors construct a test to examine LLM's ability to understand certain types of conversational implicatures by asking the models to convert non-direct answer to yes/no questions to yes/no. They found that intractable GPT-3 models perform much better than other baselines they compared but still a lot worse than human performance. They also showed that scaling doesn't help on this task. In-context learning helps the task up to 5 examples.

**Summary Of The Review:**

Good paper on analyzing implicature resolution of LLMs.

---

> ### Author Response · Authors · 2022-11-10
> **(Part 1/1) Response to reviewer J3vy -- Point 1 & 2**
>
> We thank **reviewer J3vy** for the thoughtful and very supportive review. It’s great to read your summary saying *“Good paper on analyzing implicature resolution of LLMs”*, that *“This task and experiment design in this paper is very straightforward, and it is presented clearly”*, and *“The experiments make sense and the results are informative”*. We hope to address your two points of constructive feedback below, and discuss how this has helped us improve the paper in response (a full revision has been uploaded as well as a pdf with the revisions highlighted in red in the supplementary material folder).
>
> ### Point 1 -- uncertainty in Figure 3 & 4
> *Point 1*: *“Given the small size of the dataset, some of the results for smaller categories may not be significant. It would be nice if the authors could calculate the uncertainty for Figure 3 & 4.”*. Thanks for pointing this out -- it’s a good point. While producing Figure 3 (the breakdown of accuracy for different types of examples in the dataset) we indeed found that for a few of the labels the pattern shown in Figure 3 is not significant enough to say anything meaningful about. This is why in the text we only discuss the significant effects: type labels “particularised” and “generalised”. That said, we have added the following sentence to the caption of Figure 3 in the uploaded revision to clarify this: *“The type labels World knowledge, Idiom, and Rhetorical question do not show a significantly meaningful pattern.”*
> The reason we did not add error bars to the original Figure 3 is that the relative nature of the Y-axis in the plot (which shows how label particularised and generalised drift away from the mean) doesn’t allow for it. For clarity, we have added a new plot to the manuscript in the appendix in Section F.4. This one instead shows absolute accuracy and just the Particularised and Generalised labels (and the mean) **containing the error bars to show uncertainty**, showing that this pattern is likely meaningful.
>
> For Figure 4, each line is the mean accuracy of 600 test examples. Because this plot shows one line per run through the model (prompt template), we cannot show error bars for this particular figure. Apart from showing that different models react differently to increased k in the k-shot prompt for different types of prompt templates (natural and structured) we do not make any conclusions on the significance of the difference between lines.
>
> ### Point 2 -- Title
> *Point 2*: *“The title is too eye-catching-oriented and a little disconnected to the main claim of this paper. [...] Communication is a much broader concepts than implicature. Not being able to understand implicature than average humans doesn't mean LLMs cannot communicate.”*. For a response to this point, kindly refer to the [common message about the title to all reviewers above](https://openreview.net/forum?id=WgbcOQMNXB&noteId=UcFHII1IXi). We explain our rationale for the title, but accept a change is warranted and suggest an alternative we would love to get your feedback on.
>
> ### Concluding remarks
>
> We hope the reviewer’s points of weakness are adequately addressed by the additional plot showing the uncertainty for Figure 3, the accompanying changes to the revised manuscript, and the title change. If not, we would be grateful if you could indicate outstanding points of concern which stand in the way of you raising your score, and we will happily seek to address them.

---

### Official Review · Reviewer_uf2H · 2022-10-26

**Confidence:** 4
**Correctness:** 2
**Technical Novelty And Significance:** 2
**Empirical Novelty And Significance:** 2
**Recommendation:** 5

**Clarity, Quality, Novelty And Reproducibility:**

This paper is well written and seems reproducible (assuming the dataset is released). Small clarification questions and typos are added below.

Clarification questions:

1. Page 4. “..the order in which these examples are presented matters.” ← can the authors clarify why this is the case, since there is no explanation given.
2. On naming “natural” vs. “structured” prompts. This seems somewhat non-intuitive–can the authors clarify this naming strategy?
3. Why are the fine-tuned models called “instructable” models?

Small typos:

Page 4. “.. change in protocol allows us to control for two two sources..”



**Strength And Weaknesses:**

Strengths

1. This paper is well-motivated and the evaluation is formulated clearly and concretely.
2. The authors spend time evaluating all models in different ways (e.g., relative accuracy, comparison across model sizes and classes, measuring the effect of in-context examples and so on) which is more comprehensive than most evaluations and helps understand model performance
3. The authors report variance with the accuracy results which is good to see—however it looks like the variance is exceptionally high?
4. The authors also measure the effect of a larger number of samples (k), whether or not the wording of samples in the prompt affects performance, and they break down performance by example type which allows a nice and comprehensive analysis

Weaknesses

1. The framing of this paper (and the title) are misleading. This paper focuses on evaluating implicature, which is a very important phenomena for communication, however is not everything that is needed for (zero-shot) communication and the title implies that this paper focuses on all aspects of communication. There are other aspects of communication/intent (e.g., assertives, hedges, subjective words) and so on that are nuances aspects of communication, as well as other important pragmatic effects, and this paper does not touch upon any of those.
2. The title also does not currently contain information/the word “implicature” which is misleading, so I urge the authors to reframe/change it to remove “zero-shot communicators” (since this is not something the paper tackles) and add “evaluating implicatures” (which is what the paper is largely about)
3. There are several citations re: language models and implicature that are missing. They are: “Linguistic models for analyzing and detecting biased language. Recasens, M., Danescu-Niculescu-Mizil, C., and Jurafsky, D. (2013)”, “More than words: Syntactic packaging and implicit sentiment. ​​Greene, S. and Resnik, P. (2009)” “Was it “said” or was it “claimed”? How linguistic bias affects generative language models. Patel and Pavlick (2021)”


**Summary Of The Paper:**

This paper evaluates whether or not large language models understand implicature. They create (automatically) a dataset that allows them to evaluate pre-trained models’ understanding of this phenomenon. They evaluate LLMs pre-trained on large text corpora and also ones that are fine-tuned with RL feedback to see the differences in models.


**Summary Of The Review:**

This paper studies an interesting problem however the framing of the paper is misleading.

---

> ### Author Response · Authors · 2022-11-10
> **(Part 2/2) Response to reviewer uf2H -- clarification questions**
>
> 1. To clarify the statement that *“the order in which these examples are presented matters”*, we point to Lu et al. in ACL 2022 who empirically show that LLMs are sensitive to prompt example ordering. This means if you prompt a model with a 2-shot example prompt, which example comes first in the prompt might impact performance.
>
> 2. “natural” vs. “structured” points to the prompts either being natural language, where the implicature examples are wrapped in a template to form a natural language sentence, or structured language, where the examples are wrapped into templates that are not natural language but rather structured in a exam-question-like format.
>
> 3. Finetuned models in this case are called instructable models because they are all finetuned on tasks phrased as instructions, a trend started by Ouyang (2022) and continued with works like Sanh et al (2022), Wei et al. (2022), and Chung et al (2022). For further reading on this topic we refer the reviewer to Chung et al. (2022) who nicely define what instruction finetuning is. Finally, we note that the methodology used by OpenAI is substantially different in how it’s done and uses reinforcement learning, but is philosophically similar to these works.
> Hope that helps!
>
> Thanks for pointing out the typo!

---

> ### Author Response · Authors · 2022-11-10
> **(Part 1/2) Response to reviewer uf2H**
>
> We thank **reviewer uf2H** for taking the time to comment on our work, saying it *“studies an interesting problem”* and *“is well-motivated and the evaluation is formulated clearly and concretely”*, stating the evaluation protocol *“is more comprehensive than most evaluations and helps understand model performance”*. Below we discuss how we have taken onboard your constructive criticism and improved the paper in response (a full revision has been uploaded as well as a pdf with the revisions highlighted in red in the supplementary material folder).
>
> ### Point 1-3 -- Title and framing
>
> We believe the main reason reviewer uf2H rates our work marginally below acceptance threshold is due to the title of the paper being misleading. For this point, we kindly refer the reviewer to a [main comment to all reviewers above](https://openreview.net/forum?id=WgbcOQMNXB&noteId=UcFHII1IXi) where we propose a change of title to: **“Q: Do large language models understand implicature? A: Do pigs fly?”**. Hereby we hopefully addressed the following statement by the reviewer: *“I urge the authors to reframe/change it to remove ‘zero-shot communicators’ (since this is not something the paper tackles) and add ‘evaluating implicatures’ (which is what the paper is largely about)”* (point 2).
>
> In the rest of the paper, we do not mean to claim implicature is everything that is needed for communication, but that it is a necessary condition to understand implicature to communicate with humans. As for the point that the paper framing beyond the title is misleading (part of point 1), we hypothesise that the reviewer might be referring to the first paragraph of the introduction (if not, we would be grateful if you could let us know which parts of the paper are misleading). To get across more clearly that the paper is about one aspect of communication we added the following sentence prominently to the first paragraph of the introduction in our uploaded revision: *“Although communication encompasses much more than just implicatures, such as assertives and other illocutionary acts, we view implicature understanding as a necessary condition for communicating with humans.”*
>
> *Point 3*: thanks for pointing out these three related works to us! We added them to the uploaded revision in the background on implicature (Section B of the Appendix). To quote our revision here for easy reference: *“In the context of computational models, prior work uses insights from pragmatics to show that the use of certain words can make a language model produce biased completions (Patel & Pavlick (2021), e.g. saying someone “claimed” something rather than “said” something), and inform bias and sentiment classifiers (Greene & Resnik, 2009; Recasens et al., 2013)”*
>
> ### Concluding remarks
>
> We hope that the reviewer’s points of weakness are adequately addressed by the title change and addition to the introduction. If not, we would be grateful if you could indicate outstanding points of concern which stand in the way of you raising your score, and we will gladly seek to address them.
>
> (clarification questions response below)

---

### Official Review · Reviewer_pKwN · 2022-10-26

**Confidence:** 3
**Clarity, Quality, Novelty And Reproducibility:** These are all covered in the sections…
**Correctness:** 4
**Technical Novelty And Significance:** 3
**Empirical Novelty And Significance:** 4
**Recommendation:** 6

**Strength And Weaknesses:**

**Pros**
 - Rigorously evaluating implicature understanding is an interesting contribution to a timely and important area, namely, LLMs understanding of language and communication abilities.
 - The protocol is extremely simple. This might seem like a Con, but often times identifying simple ideas and executing them correctly requires much heavy lifting. The authors put in the work to present their idea clearly: the paper is well-written and concepts are clearly defined and demarcated.
 - Overall, the empirical evaluations are easy to understand and fairly complete (Cons 2-5, below, notwithstanding). The in-depth breakdowns shed light on the general claim that LLMs fail at zero-shot implicature understanding.
 - I commend the authors for identifying a subset of their experiments that can be run without access to paid APIs or large compute.
 - The authors survey existing literature on implicature.
 - Cons 6-8 aside, the findings of empirical evaluations are presented in a careful and complete manner. The experiments seem reproducible (though I do not have API access or compute to attempt a full reproduction).

**Cons & suggestions for improvement**
Items (6-8) refer to the code and reproducibility, and items (9-14) are minor comments.

 1. The title is too general -- it's not necessary to replace the task of implicature understanding with the much more general task of "communication". If anything, you can drop "Zero-Shot". Concretely, I'd suggest something along the lines of "Large Language Models Do Not Understand Implicature" or, less accurate but more accessible, "Large Language Models Do Not Understand Implication".
 2. The perfomance gaps of LLMs and humans are about 14% in zero-shot evaluation, which decreases to 6% in few-shot evaluation. Can you quantify what these gaps mean? How much better is 6% than 14%? Given that humans perform imperfectly in this task (86%), I imagine there is a some *x%* threshold past which perfomance is "essentially" human. How close is this threshold to $x=86$? This likely requires a new set of experiments, but seems an important step forward that I would suggest as future work. For example, a Turing test in which a distinguisher prompts a black box with an implicature (or sequence of these) and must distinguish an LLM from another human.
 3. In Table 1, the differences between Davinci-001 and Davinci-002 are within each others' standard deviation. Does this affect the statistical significance / validity of the claim that Davinci-001 outperforms Davinci-002 (page 6)? Not much changes in the paper if you say that the Davincis are indistinguishable, but I find this glaring and worth a brief discussion in the paper.
 4. The identity of the asker/answerer among the pairs (Esther,Juan), (Karen,William) and (Bob,Alice) may affect the performance of the models: for example, a question asked by a Karen may be interpreted differently than a question asked by a William. I suggest adding a small ablation permuting the order of the pairs to test this.
 5. Why were human subjects only shown Prompt #2, rather than a mixture of all six prompts? As the authors find, structured prompts are easier for LLMs than natural ones. It is likely too late to test the effect of multiple vs. single prompt types on human performance, but I would expect to find an explicit discussion of this choice in the text.
 6. **Code and reproducibility.** No Python version is specified in the readme, and the code is incompatible with any version other than the intended one. Which version should be used for reproducing the results? I managed to infer that it’s 3.9 using binary search:
    - Your use of the walrus operator (’:=’) implies Python≥3.8.
    - Your requirement of torch==1.10.2 implies Python<3.10.
    - Your use of union (’|’) operator implies Python≥3.9.
 7. Even after finding the right Python version, the code does not run out of the box. Specifically, I had to manually downgrade protobuf to 3.20 (pip install -Iv protobuf==3.20). You may want to add this to your requirements.txt.
 8. You should add a lightweight and easy-to-run sanity test. For example, an argument-less script that that trains a model on tiny data, validates on the same data, and asserts very high accuracy. While the small experiment from the readme did reproduce for me, it took several minutes to run; it's good to have something that fails faster than that in case things aren't working.
 9. **Minor comments.** In the paragraph entitled "On Prompting" (page 8), the authors state that "There is a narrative around large language models that if they fail a task, it might that the prompt was not the right one." This statement should be supported by citations (just a few examples would suffice).
 10. A footnote next to a punctuation mark comes after the mark, rather than before it. In page 5, move footnote 3 to come after the period.
 11. Although Wittgenstein is cited in the appendix, I expected to find him cited alongside Grice and Huang in page 1, supporting the statement that meaning is determined by contexts, beliefs and social institutions. Not a big deal, I'll leave this decision up to the authors.
 12. Consider changing the title of Appendix B to "Background on Implicatures" or just "Implicatures".
 13. On page 19, the authors write that Bach argues that "conventional implicatures are [...] instances of something else". Can you give an example of what this something might be?
 14. It would be nice to conclude Appendix B with a paragraph explaining in what way this work is informed by the literature mentioned in this section. What is the connection between conventional/conversational implicatures (in the appendix) and generalized/particularized implicatures? If conventional/conversational implicatures do not make an appearance in the rest of the work, why the long discussion of them?

**Summary Of The Paper:**

Main contributions:
1) Identify implicature understanding as a useful benchmark of LLM's communication ability.
2) Design a protocol for comparing implicature understanding of LLMs versus that of humans.
3) Roll-out the protocol on a host of existing LLMs. Providing a deeper analysis of (a) how different properties of LLMs (e.g., instrucability or size) affect their performance, (b) which types of implicatures are harder for LLMs to understand, and (c) the effects of multi-shot learning on performance.

Broadly speaking, implicature refers to the act of saying one thing that implies another. The authors observe that any entity (or model) that is claimed to "understand language" or "be able to communicate" must, in particular, understand implicature. Continuing the wide range of works aimed at understanding and evaluating LLMs, the authors design a protocol for evaluating an LLM's ability to understand implicature.

The protocol entails prompting an LM (or human) with text of an implicature in the form of a question and answer; the task is to classify whether the answer means "yes" or "no" (logits of any token other than yes/no are masked, I think). Performing better-than-random (50%) at this task necessitates implicature understanding. The authors use six prompt templates, some *structured* (like script) and some *natural* (like prose).

Using this protocol, the authors compare the performance of various LLMs and benchmark these against humans. Overall, the authors find that the zero-shot performance of most LLMs is fairly close to random (around 60%), with the best models achieving around 72% accuracy -- compared to the mean 86% of humans. From this, the authors conclude that LLMs fail at zero-shot implicature understanding. Few-shot evaluation sees an increase to about 80%.

In addition to these main findings, the authors perform several refined anlyses:
 - breakdown by prompt type, showing that structured prompts are easier than natural prompts, but the overall ranking among models is not sensitive to prompt type
 - breakdown by implicatures type, showing that generalized implicature is easier than particularized implicature.
 - in *k*-shot evaluation, how performance scales with *k*, showing diminishing returns for $k \geq 5$.
 - how performance scales with model size (the conclusion here was unclear to me).

**Summary Of The Review:**

This paper offers a significant contribution to an important area of research. Overall, it is written well, its novelties are well-defined and explored, and the main claims are sufficiently supported by empirical findings. Do not take the many suggestions I made above as a serious negative sign; only a handful of them noticeably impact the paper. I strongly suggest the authors to make the experiments more easily reproducible by fixing Cons 6-8.

EDIT: The authors' rebuttal adequately addresses my concerns (primarily reproducibility and the title) and I have increased my score.

---

> ### Author Response · Authors · 2022-11-10
> **(part 1/3) Response to reviewer pKwN  -- points 1-5**
>
> We thank **reviewer pKwN** for the thoughtful, supportive, and in-depth review. We are happy that you think *“this paper offers a significant contribution to an important area of research. Overall, it is written well, its novelties are well-defined and explored, and the main claims are sufficiently supported by empirical findings”*. We take onboard your constructive criticism, and discuss below how we have improved the paper in response (a full revision has been uploaded as well as a pdf with the revisions highlighted in red in the supplementary material folder).
>
> ### Response to 1-5 -- Title, meaning of performance gaps, asker/answerer swapped, and human experiment
>
> *Point 1*: please refer to the [common message](https://openreview.net/forum?id=WgbcOQMNXB&noteId=UcFHII1IXi) about the title to all reviewers above. We explain our rationale for the title, but accept a change is warranted and suggest an alternative we would love to get your feedback on.
>
> *Point 2*: You raise an interesting point saying: *“The perfomance gaps of LLMs and humans are about 14% in zero-shot evaluation, which decreases to 6% in few-shot evaluation. Can you quantify what these gaps mean?”* and *“I imagine there is a some x% threshold past which perfomance is ‘essentially’ human. How close is this threshold to x=86?”*. Implicature resolution is an ambiguous task, different context interpretations will lead to different implicatures. This is one of the reasons humans perform imperfectly at 86%. Your suggestion of evaluating the human-machine performance gap by a type of Turing test where a human tries to distinguish the model from the human by a sequence of implicature prompts is an interesting and important future work avenue.
> In our current work we can turn to three results to quantify whether this gap is meaningful. Firstly, the high inter-annotator agreement points to a large portion of the data in this benchmark not being ambiguous to humans. Secondly, the worst score humans get on the test set is 83.5%. Finally, the st. dev. is 2.3% among humans. This all points to a gap of 14% and probably even 6% being significant. Additionally, it points to a 6% gap being significantly better than a 14% gap. Your future work suggestion is a great addition that we fitted in the conclusion of the revision, copied here for convenience: *“Additionally, an interesting question for future work is for which accuracy models will be indistinguishable from humans. This could be answered with a type of Turing test in which a human must distinguish a LLM from another human by prompting both with a sequence of implicatures.”*
>
> *Point 3*: *“In Table 1, the differences between Davinci-001 and Davinci-002 are within each others' standard deviation.”* That is a good point, and indeed when considering different wordings of prompt templates the picture of Davinci-1 being better than 2 zero-shot might change. We have reworded the paragraph **The best performing model classes overall** in Section 4.1 to reflect this (see uploaded revision). A small snippet of that revision here: *“The OpenAI models perform significantly better than any other. [...] but the difference with InstructGPT-3-175B is not significant.”*
>
> *Point 4*: *“The identity of the asker/answerer among the pairs (Esther,Juan), (Karen,William) and (Bob,Alice) may affect the performance of the models”* That’s an interesting suggestion, although we hypothesise that the fact that we already have three different templates that only differ in names would mitigate the effect of this. Nonetheless, we ran an experiment on three big models, Cohere-XL, Davinci-001, and Davinci-002, to verify. We use the natural templates, but swap the asker and answerer names. The zero-shot results for these new templates are 63.39 +/- 0.83, 71.2% +/- 1.5, and 68.06 +/- 0.67. Comparing these to the old natural templates scores for these models 62.4% +/- 0.3, 71.5% +/- 1.1, and 68.5% +/- 0.8 (taken from Table 1), they don’t differ much. As can be seen in Appendix F.3, the API stochasticity can mostly account for the small variation. We are in the process of running the same experiment on larger models that are not behind an API, but this takes a bit longer. We will add this experiment to the appendix when finalised!
>
> *Point 5*: *“Why were human subjects only shown Prompt #2, rather than a mixture of all six prompts?”* We have added a discussion for this design choice in the appendix section on the human experiment design (Section E in the uploaded revision). To summarise this discussion here: although models have been shown to be very sensitive to prompt wording, we are not aware of any such studies showing humans resolve implicatures differently based on the way they are presented. For this reason, we opted to have a full labeled set on Prompt #2 rather than a mixture of all prompts. That said, to confirm this hypothesis future work could investigate the effect of different wordings on implicature resolution by humans.

---

> ### Author Response · Authors · 2022-11-10
> **(part 2/3) Response to reviewer pKwN -- points 6-8 on code and reproducibility**
>
> Thanks for taking the time to improve our code, you did not need to do this (a reviewer should never have to perform binary search to find out the Python version needed to run it) and it’s a major aspect of reproducibility in our field! We have added a section to the README headed “Check installation” containing the Python version (*point 6*, indeed your bin. search was effective, it is Python 3.9.10), as well as the necessary prelims to install (*point 7*, like rust for the `transformers` library). We added a folder of tests and additionally added a lightweight end-to-end run test (`code/test_code_runs.sh`) that runs in << a minute (*point 8*).

---

> ### Author Response · Authors · 2022-11-10
> **(part 3/3) Response to reviewer pKwN -- minor comments 9-14**
>
> ### Minor comments 9-14 -- clarifications & writing
> Thanks for these valuable suggestions, we incorporated the writing changes and citations in the revision. On point 13 and 14 (*“conventional implicatures are [...] instances of something else". Can you give an example of what this something might be? What is the connection between conventional/conversational implicatures (in the appendix) and generalized/particularized implicatures?”*); we rewrote the background section to point to one of the semantic concepts Bach classifies conventional implicatures under (namely “utterance modifiers”), and paragraph “Breaking down performance per example type” in Section 4.1 to correct a confusion on the relation generalised/particularised to conventional/conversation. In short; the demarcation we adhere to is Gricean. Conventional implicatures, generalized conversational implicatures, and particularised conversational implicatures. Conventional implicatures arise due to the conventional meaning of the word (“therefore” implying a causal relation) and are importantly not cancellable without creating incoherence, generalized implicatures are cancellable but require little or no context to be resolved (“some” implying “not all” in certain contexts), and particularised implicatures are cancellable and arise only in specific contexts (“want to say for a nightcap?”, “gotta get up early”). For more details please refer to the last paragraph of the background Section B in the appendix.
>
> ### Concluding remarks
> We thank you again for your valuable feedback and constructive criticism, which we hope we have adequately addressed and/or incorporated as per the comments and replies above. Please let us know if there are any outstanding concerns that would inhibit you from raising your score, and we will gladly look into addressing them.

---

> ### Comment · Reviewer_pKwN · 2022-12-12
> **Final comment**
>
> Dear area chair and authors,
>
> After reading the discussions in this forum, as well as a virtual meeting with fellow reviewers, I am writing this comment to finalize my review. In the meeting, fellow reviewers made two good points regarding this paper: (1) human performance in the evaluated task does not exceed 90%; and (2) a recent LLM exhibits a "phase-transition" in performance, and we might expect to see a similar phenomenon in the implicature task if sufficiently large models are used.
>
> These are both good points. Point (1) is addressed in the discussion I had with the authors and subsequently incorporated into the paper as future work. Point (2) has been discussed in the recent discussion between the authors and the area chair. The authors point out that the referenced benchmark is not peer-reviewed, and that the model exhibiting phase-transition (PaLM) is not openly accessible. In my view, this puts this point in scientific limbo: not (yet) reviewed by peers, and not immediately addressable by the authors. Although the authors could have scaled up the openly-accessible LMs used in their own work to explore the possibility of phase-transition, it is unclear whether a negative answer (no phase-transition; LMs still fail at implicature) would even have refuted the hypothesis of phase-transition, given the different architectures. I encourage the authors to incorporate this discussion as an avenue of future work, but ultimately I decided not to lower my score as a result of this unexplored direction.
>
> The score given in my review (8) reflects my current opinion of this paper. Thank you authors, reviewers and AC for the interesting and earnest discussion.

---

> > ### Comment · Reviewer_pKwN · 2022-12-12
> > **Re: Final comment**
> >
> > Update regarding BIG benchmark question: I'd like to direct everyone's attention to the [last point](https://iclr.cc/Conferences/2023/ReviewerGuide) in ICLR23's reviewer guidelines, per which reviewers are expected to use "their own good judgement" regarding whether authors are expected to know about / cite non-peer-reviewed papers.
> >
> > It seems to me that BIG benchmark is fairly well known in the community. More importantly, the AC has brought to my attention the fact that the authors have already cited the BIG benchmark paper in their work: "The standard set of benchmarks LLMs are further evaluated on covers tasks like question answering ... and more [Srivastava et al. 2022]. ... none of these benchmarks explicitly evaluate implicature understanding."
> >
> > The second part of this quote leaves me puzzled, as implicatures are clearly listed as one of the evaluation tasks in BIG ([section 7.2](https://arxiv.org/pdf/2206.04615.pdf)). This error directly undermines the main claim made in this paper: the claim is that LLMs do not understand implicatures, yet a cited paper points towards evidence that they do. Due to this, I am lowering my score back to 6.
> >
> > Though this may be disappointing, I hope the authors are not too discouraged by this update of mine. Indeed, I still think the work done in this paper is a useful contribution to the literature. A constructive takeaway I can offer from our discussions is that perhaps this work should be viewed as proposing and a useful benchmark for testing implicature-understanding in LLMs, rather than concluding that LLMs do not understand implicatures.

---

> > > ### Author Response · Authors · 2022-12-13
> > > **BIG bench task does not mean LLMs understand implicature**
> > >
> > > We thank the reviewer for their response and engagement with this discussion. We would like to briefly point out the part of our initial response to the AC where we mention Section 7.2; the task is indeed listed in the list of all BIG bench tasks, but not mentioned anywhere else. There is no dedicated discussion section about implicatures or the implicature benchmark result. We did not notice the entry in the list at the end of the paper when reading the BIG bench paper.
> > >
> > > About the point that this means LLMs do understand implicature, we respectfully disagree that this can be concluded from the result on this task. As also said in the comment to the AC. The phase transitions by PaLM are exclusively with 3- and 4-shot prompts. Additionally, the accuracy of their task and ours is incomparable because they preprocess the data differently than we do. Implicatures are inherently ambiguous utterances, and the way we handle that is different. The contributors of the implicatures BIG bench task throw out data they themselves deem ambiguous, whereas we implicitly let a set of human annotators decide what is ambiguous by virtue of their inter-annotator agreement on these data. As a result, the best human performance on the BIG bench task is 100%, whereas our human best performance is closer to 90%. PaLM surpasses their human average of 80%, which seems a very low human average given that 100% can be reached (it's lower than our human performance even though we _do_ keep the ambiguous data). In light of these differences, we expect few-shot PaLM to not surpass human performance on our task, but we have no way of evaluating this as PaLM is not available to researchers outside of Google. We would be very interested in evaluating PaLM on our task.

---

> > ### Author Response · Authors · 2022-12-13
> > **Clarification question**
> >
> > Maybe we're misreading this, but are you intending to suggest that we train larger language models than are currently publicly available? We evaluate on multiple OpenAI models, OPT, and BLOOM. These models are all 175B parameters (or are believed to be, in the case of OpenAI’s models), far smaller than PaLM’s 540B parameter model. We are unaware of any publicly available dense models anywhere close to 540B parameters.

---

### Author Response · Authors · 2022-11-10
**A note to all reviewers about the changed title of this paper**

We sincerely thank all reviewers for pointing out to us that the title of this work is perceived as too general for what the paper in fact tackles; communication encompasses much more than just implicatures. We have decided to act on the signal that all four reviewers independently have issues with the title, and hope to get your thoughts on the following title suggestion: **“Q: Do large language models understand implicature? A: Do pigs fly?”**. For the sake of clarity, we want to explain to the reviewers the reasoning behind the original title, hoping this explanation (and the further changes to the introduction described within) help justify our framing of the thesis this paper presents.

**Reason for original title**

The reason for choosing the original title “Large language models are not zero-shot communicators” was that the authors collectively view implicature as a necessary, but not sufficient, condition for communication. We agree that there are many more aspects to communication than just implicature, but we hold the view that one cannot communicate effectively with humans if language is not understood in context. To quote G.M. Green in her introduction to pragmatics textbook titled *“Pragmatics and Natural Language Understanding”: “the term communication presupposes achievement of the intended effect of verbal action upon the addressee [...]. Communication is [...] the successful interpretation by an addressee of a speaker’s intent in performing a linguistic act.”.*

Given that this did not come across clearly in the paper, we have added the following line to the first paragraph in the introduction (see uploaded revision): *“Although communication encompasses much more than just implicatures, such as assertives and other illocutionary acts, we view implicature understanding as a necessary condition for communicating with humans.”* That said, we hear the reviewers, and we are more than happy to change the title.

---

### Author Response · Authors · 2022-11-16
**To all reviewers; deadline for revisions 18th November**

Dear reviewers,

Although we believe we have significantly improved our submission in response to your comments and suggestions (which can be seen in the uploaded pdf or in the red-coded revision in the supplementary material), we would like to kindly ask if you could verify whether you think your reviews have been adequately addressed before the 18th of November (marking the ending of discussion stage 1). After the 18th discussion stage 2 begins and we cannot update our submission anymore. In the case of remaining comments requiring revisions, we will gladly seek to address them.

---

### Author Response · Authors · 2022-12-07
**To all reviewers; we kindly ask for a response to our rebuttal (deadline December 12)**

Dear Reviewers,

Thank you for taking the time to review our work, and thanks to reviewer pKwN for increasing their score to an 8. To all other reviewers, we have carefully considered your comments and have provided a thorough rebuttal addressing your concerns. If you feel that your comments have been adequately addressed, we would greatly appreciate it if you could update your score to reflect that. Thank you again for your valuable input.

---

### Comment · Area_Chair_ABcG · 2022-12-08
**Question about related work**

I noticed a very related implicatures task in the [Big bench](https://github.com/google/BIG-bench/tree/main/bigbench/benchmark_tasks/implicatures) benchmark that was cited in this paper. Interestingly, one can see that there are some surprising phase transitions as the PaLM model grows to 540B, something that has been claimed to be a notable feature of massive language models. I am curious how this relates to the current work.

---

> ### Author Response · Authors · 2022-12-09
> **Thanks for pointing out this related BIG bench task**
>
> We thank the Area Chair for pointing out this related task to us. It appears to be very close to the dataset contribution of our work. We did significant due diligence when it came to prior art, as is hopefully evidenced by our related work section. In doing so we primarily focussed on peer reviewed work, which the Google implicature dataset is not. Additionally, this task is not explicitly treated in the BIG bench technical report. It is only mentioned in the comprehensive list of all 171 tasks at the end of the report. The report has no analysis or discussion of the majority of the tasks. As a result, we believe one could be forgiven for overlooking a related contribution nested several folders deep in a large collection of benchmarks. Of course, the task is very related to ours and we are grateful for the AC to have pointed it out to us.
>
> That being said, we believe the strength of our contribution is still clear. Our paper proposes a reproducible annotation protocol and evaluates it against contemporary and publicly accessible models, discusses in depth the need for such pragmatic evaluation and situates the contribution in contemporary literature on LLMs, and finally provides an extensive analysis of the results along with that of auxiliary experiments (e.g. few shot learning and prompt-sensitivity). These contributions go above and beyond Google's BIG Bench contribution in both justifying the need for implicature-based evaluation of pragmatics, and providing insight into what class of models perform well. The authors of the BIG Bench paper did not evaluate on publicly available language models and consequently have no models in common with our paper. Note that the "GPT-3" models evaluated in that paper appear to be the original models trained for the paper "Language Models are Few Shot Learners," while ours are the models that are served in the OpenAI API. While we hope that these models are similar, experts in evaluating large language models believe they are not the same.
>
> The phase transitions the AC refers to appear to apply exclusively to PaLM with 3- and 4-shot prompts. It is certainly possible that this emergent performance would show up on our benchmark, but we have no way of knowing. We see no evidence of emergent properties in any of our evaluations, and frequently see non-standard scaling curves. Additionally, the accuracy of their task and ours is incomparable because they preprocess the data differently than we do. Implicatures are inherently ambiguous utterances, and the way we handle that is different. The contributors of the implicatures BIG bench task throw out data they themselves deem ambiguous, whereas we implicitly let a set of human annotators decide what is ambiguous by virtue of their inter-annotator agreement on these data. As a result, the best human performance on the BIG bench task is 100%, whereas our human best performance is closer to 90%. In light of these differences, we expect few-shot PaLM to not surpass human performance on our task, but we have no way of evaluating this as PaLM is not available to researchers outside of Google. We would be very interested in evaluating PaLM on our task.
>
> In terms of incorporating this new information into the paper, we would of course first and foremost reference the BIG-bench repository as the first work to adapt the data of George & Mamidi (2020), and as having independently made a powerful argument for the pursuit of pragmatics-based evaluations. In our experimental section, we can reference their results in general, and in particular their results on PaLM, noting any differences in evaluation protocol and, in passing, the community's inability to reproduce such results externally or on related/future benchmarks or experimental evaluation protocols.

---

### Public Comment · ~Liang_Chen10 · 2023-02-28
**Question about evaluation of OpenAI models.**

Hi, Thanks you for interesting work!

I wonder that how do you get the output probability of models that are only provided with web API, like the instruct-gpt3. My APIs only respond with most possible texts output.

Thanks~

---

> ### Author Response · Authors · 2023-02-28
> **Output probability with OpenAI API**
>
> Hello,
>
> Thanks for your comment! In the supplementary work folder you can see the code in a folder `code`. In the file `code/src/models.py` at line 329 you can see the function to get the likelihood scores with the OpenAI API.
>
> Basically the only thing you need to do is set `logprobs=1` when you call `openai.Completion.create()`
>
> Hope that helps!

---

### Decision · Program_Chairs · 2023-01-20

**Decision:**

Reject

**Justification For Why Not Higher Score:**

Two major problems: 1. The well-known Big-Bench has a 14-month-old LLM benchmark on exactly the same task, using much larger language models, that the authors were unaware of, even though they themselves cite it. 2. The framing of the paper is sensational and unscientific.

**Justification For Why Not Lower Score:**

N/A

**Metareview: Summary, Strengths And Weaknesses:**

This is an interesting paper about what large language models understand. The focus is "implicatures," an important aspect of communication. For example, if you ask if "Did you take out the trash?" and someone says "All 12 bins" the implication is "Yes." The present paper designs a protocol and evaluates several language models on their understanding of implicatures both zero-shot and above.

The reviewers liked this question and found the execution reasonable but felt that the overall framing and discussion of related work was inadequate. Also, there were concerns about the inter-rater reliability though these were less significant.

In particular, the original submission title "Large language models are not zero-shot communicators" makes a overly strong claim not convincingly established in the paper, and is at the same time not specific to what this paper is about. Furthermore, it is not clear which language models are being referred to: all language models, past, present and future? all present language models? all language models that the authors tried out? The authors propose changing the title to a cute new title that focuses on implicatures, which is good, but it is still carries a tone that makes a vague claim about the ability of (all?) large language models. Important to the decision was the fact that one of the papers that the authors cited a well-known paper that pointed to 14-month-old benchmark for implicatures on large language models. In fact, while the benchmark itself has not yet been peer reviewed, the data it has claims that a few-shot 540B language model outperforms average humans. Whether or not this is true, its existence highlights the problems with the current paper's framing and claims that it is unlikely that scaling will work.

The authors may consider reframing the paper as discussing the *evaluation* of language models on the axis of implicatures, and what is novel/interesting about that.

**Summary Of Ac-Reviewer Meeting:**

There was agreement that the current framing is too strong and somewhat sensational. The meeting prompted the AC to do a simple google search which uncovered the prior benchmark (which the authors themselves had cited but apparently hadn't bothered to do a simple search), which led to further discussion between the AC, authors, and reviewers on the forum. Ultimately scores were lowered.

---

> ### Author Response · Authors · 2023-01-27
> **Thanks for all the feedback + pushback on an unfair assumption**
>
> Thank you to the AC and the reviewers for all their feedback.
>
> We would like to highlight the assumption that we "apparently hadn't bothered to do a simple search", to be counter-productive. We appreciate that the review process is adversarial, however to suggest we lack integrity and did not try our best to find all the related works is unfair, as our extensive related work section shows. We genuinely missed this subtask within BIG bench even though we cited the BIG bench report; the task is one of 200+ tasks and not treated in the report except as part of a list of all tasks. We take full responsibility for missing this and accept the rejection on these grounds.
>
> Most further points in this comment are already addressed in [our comment below](https://openreview.net/forum?id=WgbcOQMNXB&noteId=zHPUdaD0AUK).